# Strategic Data Sharing between Competitors

**Nikita Tsoy**
INSAIT, Sofia University
Sofia, Bulgaria
`nikita.tsoy@insait.ai`

**Nikola Konstantinov**
INSAIT, Sofia University
Sofia, Bulgaria
`nikola.konstantinov@insait.ai`

## Abstract

Collaborative learning techniques have significantly advanced in recent years, enabling private model training across multiple organizations. Despite this opportunity, firms face a dilemma when considering data sharing with competitors—while collaboration can improve a company's machine learning model, it may also benefit competitors and hence reduce profits. In this work, we introduce a general framework for analyzing this data-sharing trade-off. The framework consists of three components, representing the firms' production decisions, the effect of additional data on model quality, and the data-sharing negotiation process, respectively. We then study an instantiation of the framework, based on a conventional market model from economic theory, to identify key factors that affect collaboration incentives. Our findings indicate a profound impact of market conditions on the data-sharing incentives. In particular, we find that reduced competition, in terms of the similarities between the firms' products, and harder learning tasks foster collaboration.

## 1 Introduction

Machine learning has become integral to numerous business functions in the past decade, such as service operations optimization, the creation of new products, consumer service analytics, and risk analysis (Chui et al., 2022). Despite the transformative power of machine learning, its efficacy hinges significantly on the quality and quantity of the training data, making data a key asset corporations can use to increase their profit.

One way to enhance data access and machine learning models is collaboration via data sharing (Rieke et al., 2020; Durrant et al., 2022). Data-sharing schemes can bring benefits in many industries, such as agriculture, finance, and healthcare (Durrant et al., 2022; FedAI; Rieke et al., 2020). However, at least two barriers exist to such collaborations. The first is privacy and the associated regulatory concerns, which can be partially addressed by new collaborative learning techniques, such as federated learning (Kairouz et al., 2021). The second is proper incentives for collaboration. If the entities have no such incentives, they may not collaborate at all, free-ride (Blum et al., 2021; Karimireddy et al., 2022), or attack the shared model (Blanchard et al., 2017).

Such conflicting incentives arise naturally between market competitors. On the one hand, the competitors are an appealing source of data, since they operate on the same market and have data of a similar type. On the other hand, the collaboration could also strengthen the competitors' machine learning models, potentially leading to downstream losses for the company. This concern is especially important for big firms that can influence prices and act strategically by considering how their decisions affect competitors' responses. Strategic actions might increase profits through various channels and produce complex downstream effects. For example, firms might collude to capture

more revenue[1] or engage in a price war to win a market.[2] Thus, *data sharing between big firms might have a complicated downstream effect on their profits*.

**Our contributions**    Despite the significance of this data-sharing trade-off, particularly for large, tech-savvy companies, there is limited research on how market competition affects collaborative learning incentives. Our work aims to fill this gap by proposing a framework to investigate data-sharing incentives. The framework is modular and consists of three components: market model, data impact, and collaboration scheme. These components represent the firms' production decisions, the effect of additional data on model quality and the data-sharing negotiation process, respectively.

To investigate the key factors that influence data-sharing decisions, we instantiate our framework using a conventional market model from economic theory and a data impact model grounded in learning theory. We examine three potential collaboration schemes: binary collaboration between two firms, partial collaboration between two firms, and binary collaboration among multiple firms. Our findings consistently indicate a profound impact of market conditions on the collaboration incentives. In the case of two firms, we theoretically demonstrate that collaboration becomes more appealing as market competition, measured by the similarities of the firms' products, decreases or the learning task complexity increases. For multiple firms, we conduct simulation studies that reveal similar trends.

## 2   Related work

**Data sharing incentives in machine learning**    Incentives for collaboration in machine learning constitute an important research topic, especially in the context of new learning paradigms, such as federated learning (Kairouz et al., 2021). Some notable lines of work concern incentives under data heterogeneity (Donahue & Kleinberg, 2021; Werner et al., 2022); compensation for computation, energy, and other costs related to training (Yu et al., 2020; Tu et al., 2022; Liu et al., 2022); fair distribution of the benefits resulting from collaboration (Lyu et al., 2020a,b; Blum et al., 2021); and free-riding (Richardson et al., 2020; Karimireddy et al., 2022). We refer to Zeng et al. (2021); Yang et al. (2020); Zhan et al. (2021) for a detailed overview.

A fundamental difference between our work and the research described above is that we explicitly consider the downstream market incentives of the firms and their effects on data-sharing decisions. These incentives are orthogonal to those previously studied in the literature. For instance, the data-collection game of Karimireddy et al. (2022) can be included in our framework as an additional stage of the game after our coalition formation stage. In addition, our framework is not specific to federated learning, as it can be applied regardless of the learning procedure used on the shared dataset.

To our awareness, Wu & Yu (2022) is the only work considering data sharing between market competitors. However, the authors do not explain the mechanisms behind the effects of ML models' quality on the market. Therefore, their framework can not predict whether a potential coalition will be beneficial for the company but only indicate the benefits of the coalition post factum. In addition, they use the marketing model of Rust & Zahorik (1993) instead of the classic economic models (Tirole, 1988) to model competition, potentially constraining the applicability. Finally, their notion of the sustainability of coalitions does not arise from standard concepts in game theory. In contrast, we base our analysis on the standard Nash equilibrium concept (Nash, 1951).

**Market competition and information sharing**    Competitive behavior is a well-studied topic in the economic literature (Tirole, 1988). We refer to Shapiro (1989) for an extensive review of the theory of competition. From a more applied perspective, Reiss & Wolak (2007) discuss how to test the behavior of firms empirically, while Berry et al. (2019) overview the modern empirical studies of industrial organization. A rich economic literature studies the incentives of competing firms to share information (Raith, 1996; Bergemann & Bonatti, 2019). For example, demand (Vives, 1984) or costs (Fried, 1984) might be unobservable, and firms might share information about them to improve their decisions. In contrast to these studies, in our model, firms share information to improve their products or optimize their costs, not to reduce uncertainty. Thus, the mechanisms and conclusions of these earlier works and our work differ.

---

[1] https://www.nytimes.com/2021/01/17/technology/google-facebook-ad-deal-antitrust.html

[2] https://www.reuters.com/article/us-uber-results-breakingviews-idUSKCN1UY2X5

# 3 Data-sharing problem

In this section, we provide a general formulation of the data-sharing problem, consisting of three components: market model, data impact, and collaboration scheme. The market model outlines consumers' and firms' consumption and production decisions. Data impact explains how additional data affects machine learning model quality. Lastly, the collaboration scheme details the data-sharing negotiation process among the competitors. Given a specific application, these three components can be derived/modeled by market research or sales teams, operation management and data scientists, and mediators, respectively.

We begin with an illustrative example that will help us to clarify the abstract concepts used in our framework and then introduce the three framework components in their full generality.

## 3.1 Running example

Consider a city with a taxi market dominated by a few firms. Each firm collects data (e.g., demand for taxis and traffic situation) to train a machine learning model for optimizing driver scheduling and other internal processes, which is crucial for improving scheduling quality, reducing costs, or enhancing services. Each company can use only its own data or take part in a collaborative training procedure, like federated learning, with its competitors. Collaboration can substantially improve its machine learning model, but it may also strengthen the models of its competitors. Thus, the company must carefully evaluate the impact of data-sharing on its downstream profits.

## 3.2 Market model

The market model encompasses consumer actions (demand factors) and firm production actions (supply factors). We consider a market with $m$ firms, $F_1, \dots, F_m$, each producing $q_i \in \mathbb{R}_+$ units of good $G_i$ and offering them to consumers at price $p_i \in \mathbb{R}_+$, where $\mathbb{R}_+ = [0, \infty)$. In our example, $G_1, \dots, G_m$ represent taxi services from different companies, with consumers being city travelers, $q_i$ are the total number of kilometers serviced by company $i$, and $p_i$ are the prices per kilometer.

In this setting, prices and quantities are $2m$ unknown variables. Correspondingly, we need $2m$ constraints to describe the consumers' and firms' market decisions, which we derive from the consumers' and the firms' rationality. In contrast to standard market models, our framework allows product utilities and costs to depend on the qualities of the machine learning models of the firms $\boldsymbol{v} = (v_1, \dots, v_m)$. This dependence models the impact of the learned models on the services and production pipelines of the firms.

**Consumers' behavior**   Each consumer $j$ optimizes their utility $u^j(\boldsymbol{g}^j, \boldsymbol{q}^j, \boldsymbol{v})$ by purchasing goods given the market prices. We assume that consumers cannot influence prices, as there are many consumers and they do not cooperate.

The consumer's utility depends on three factors. The first one is the consumed quantities $\boldsymbol{q}^j = (q_1^j, \dots, q_m^j)$ of products $G_1, \dots, G_m$. The second is the machine learning models' qualities $\boldsymbol{v}$, as these may impact the utility of the corresponding products. The last one is consumed quantities $\boldsymbol{g}^j = (g_1^j, \dots, g_k^j)$ of goods outside the considered market (e.g., consumed food in our taxi example). Assuming that each consumer $j$ can only spend a certain budget $B^j$ on all goods, one obtains the following optimization problem

$$\max_{\boldsymbol{g}^j, \boldsymbol{q}^j} u^j(\boldsymbol{g}^j, \boldsymbol{q}^j, \boldsymbol{v}) \text{ s.t. } \sum_{l=1}^{k} \tilde{p}_l g_l^j + \sum_{i=1}^{m} p_i q_i^j \leq B^j, \tag{1}$$

where $\tilde{\boldsymbol{p}}$ are the outside products' prices (which we consider fixed). The solution to this problem $q_i^{j,*}(\boldsymbol{p}, \boldsymbol{v})$ determines the aggregate demanded quantity of goods

$$q_i(\boldsymbol{p}, \boldsymbol{v}) := \sum_j q_i^{j,*}(\boldsymbol{p}, \boldsymbol{v}). \tag{2}$$

The functions $q_i(\boldsymbol{p}, \boldsymbol{v})$ (known as the demand equations) link the prices $p_i$ and the demanded quantities $q_i$ and constitute the first $m$ restrictions in our setting.

**Firms' behavior**   Firms maximize their expected profits, the difference between revenue and cost,

$$\Pi_i^e = \mathbf{E}_{\boldsymbol{v}}(p_i q_i - C_i(q_i, v_i)). \tag{3}$$

Here $C_i(q_i, v_i)$ is the cost of producing $q_i$ units of good for the firm $F_i$, and the expectation is taken over the randomness in the models' quality, influenced by the dataset and the training procedure. Note that the model quality is often observed only after testing the model in production (at test time). Therefore, we assume that the firms reason in expectation instead. In our running example, $C_i$ depends on driver wages, gasoline prices, and scheduling quality.

The firms may act by either deciding on their produced quantities, resulting in what is known as the Cournot competition model (Cournot, 1838); or on their prices, resulting in the Bertrand competition model (Bertrand, 1883). As the demand equations (2) may interrelate prices and quantities for various products, firms strategically consider their competitors' actions, resulting in a Nash equilibrium. The equilibrium conditions provide another $m$ constraints, enabling us to solve the market model entirely.

### 3.3   The impact of data on the market

Each company $F_i$ possesses a dataset $D_i$ that can be used to train a machine learning model (e.g., trip data in the taxi example) and may opt to participate in a data-sharing agreement with some other firms. Denote the final dataset the company acquired by $D_i^c$. The company then uses the dataset to train a machine learning model. We *postulate two natural ways that the model quality $v_i$ can impact a company*. The first one is by *reducing the company's production costs* $C_i(q_i, v_i)$, for example, by minimizing time in traffic jams. The second is by *increasing the utility of its products* $q_i(\boldsymbol{p}, \boldsymbol{v})$, for example, by minimizing the waiting time for taxi arrival.

### 3.4   Collaboration scheme

Following classic economic logic, we posit that *firms will share data if this decision increases their expected profits $\Pi_i^e$*. Since firms can not evaluate the gains from unknown data, we assume that they know about the dataset characteristics of their competitors (e.g., their number of samples and distributional information). Although expected profit maximization determines individual data-sharing incentives, forming a coalition necessitates mutual agreement. Therefore, the precise game-theoretic formulation of the data-sharing problem depends on the negotiation process details, such as the number of participants and full or partial data sharing among firms.

Consider our taxi example. If only two firms are present, it is natural that they will agree on sharing their full datasets with each other if and only if they both expect that this will lead to increased profits. Partial data-sharing will complicate the process, leading to an intricate bargaining process between the firms. Finally, if many companies are present, the data-sharing decisions become even more complicated, as a firm needs to reason not only about the data-sharing benefits but also about the data-sharing decisions of other firms.

**Legal and training costs considerations**   Other considerations can also enter into our framework through the collaboration scheme. In particular, if firms have legal requirements on data usage, they should impose suitable constraints on possible data-sharing agreements. If the collaborative learning procedure involves training large models, the firms should negotiate how to split training costs. For example, they might split the costs equally among all coalition members or proportionally to the sizes of their datasets.

## 4   Example market and data impact models

In this section, we instantiate the general framework using a conventional market model from economic theory and a natural data impact model justified by learning theory. These models allow us to reason quantitatively about the data-sharing problem in the following sections, leading to the identification of several key factors in the data-sharing trade-off. To focus solely on the data-sharing trade-off, we make several simplifying assumptions: the firms' data is homogeneous, training costs are negligible, and legal constraints are not present or are automatically fulfilled.

## 4.1 Market model

In order to make quantitative statements about the firms' actions and profits within the general model of Section 3.2, one needs to make further parametric assumptions on utilities $u^j$ and cost functions $C_i$ and specify the competition game. To this end, we use a specific utility and cost model standard in the theoretical industrial organization literature (Tirole, 1988; Carlton & Perloff, 1990). Despite its simplicity, this model effectively captures the basic factors governing market equilibrium for many problems and is often used to obtain qualitative insights about them.

### 4.1.1 Demand

We assume that, in the aggregate, the behavior of consumers (1) can be described by a representative consumer with quasi-linear quadratic utility (QQUM, Dixit 1979)

$$\max_{g,\boldsymbol{q}} u(g,\boldsymbol{q}) := \sum_{i=1}^{m} q_i - \left( \frac{\sum_i q_i^2 + 2\gamma \sum_{i>j} q_i q_j}{2} \right) + g = \boldsymbol{\iota}^\mathsf{T}\boldsymbol{q} - \frac{\boldsymbol{q}^\mathsf{T}\mathbf{G}\boldsymbol{q}}{2} + g \text{ s.t. } g + \boldsymbol{p}^\mathsf{T}\boldsymbol{q} \le B. \quad (4)$$

Here, $\boldsymbol{\iota} = (1, \ldots, 1)^\mathsf{T}$, $\mathbf{G} = (1-\gamma)\mathbf{I} + \gamma\boldsymbol{\iota}\boldsymbol{\iota}^\mathsf{T}$ and $g$ is the quantity from a single outside good.

Additionally, $\gamma \in \left( -\frac{1}{m-1}, 1 \right)$ is a measure of substitutability between each pair of goods: higher $\gamma$ corresponds to more similar goods. In our running example, $\gamma$ describes the difference in service of two taxi companies, such as the difference in the cars' quality or the location coverage. We refer to Choné & Linnemer (2019) for a detailed discussion on the QQUM model and the plausibility of assuming an aggregate consumer behavior.

In this case, the exact form of the demand equations (2) is well-known.

**Lemma 4.1** (Amir et al. 2017). *Assume that $\mathbf{G}^{-1}(\boldsymbol{\iota}-\boldsymbol{p}) > 0$ and $\boldsymbol{p}^\mathsf{T}\mathbf{G}^{-1}(\boldsymbol{\iota}-\boldsymbol{p}) \le B$. The solution to the consumer problem (4) is*

$$p_i = 1 - q_i - \gamma \sum_{j \neq i} q_j \iff q_i = \frac{1 - \gamma - p_i - \gamma(m-2)p_i + \gamma \sum_{j \neq i} p_j}{(1-\gamma)(1+\gamma(m-1))}. \quad (5)$$

The technical conditions $\mathbf{G}^{-1}(\boldsymbol{\iota}-\boldsymbol{p}) > 0$ and $\boldsymbol{p}^\mathsf{T}\mathbf{G}^{-1}(\boldsymbol{\iota}-\boldsymbol{p}) \le B$ ensure that the consumers want to buy at least a bit of every product and do not spend all of their money in the considered market, respectively. For completeness, we prove Lemma 4.1 in Appendix A.1.

### 4.1.2 Supply

We assume that the cost functions (3) are linear in the quantities

$$\Pi_i^e = \mathbf{E}(p_i q_i - c_i q_i), \quad (6)$$

where the parameter $c_i$ depends on the quality of the machine learning model. We denote $c_i^e = \mathbf{E}_{D_i^c}(c_i)$.

*Remark* 4.2. Here we assume that the quality of the machine learning model only affects the production costs, and not the utilities. However, in this market model, it is equivalent to assuming that the machine learning model affects the consumers' utility via increasing the effective quantities of the produced goods. Specifically, assume that each product has an internal quality $w_i$ and it increases the effective amount of good in the consumer's utility (4), resulting in utility $u(g, w_1 q_1, \ldots, w_m q_m)$. This model is equivalent to the model above after substituting $q_i$, $p_i$, and $c_i$ with their effective versions $\tilde{q}_i = w_i q_i$, $\tilde{p}_i = p_i/w_i$, and $\tilde{c}_i = c_i/w_i$.

We now derive the remaining constraints on the prices and costs. Note that these equilibria depend on the expected costs $c_i^e$ and hence on the quality of the machine learning models.

**Cournot competition**    Each firm acts by choosing an output level $q_i$, which determines the prices (5) and expected profits (6). The next standard lemma describes the Nash equilibrium of this game.

**Lemma 4.3.** *Assume that the expected marginal costs are equal to $c_1^e, \ldots, c_m^e$, and companies maximize their profits (6) in the Cournot competition game. If $\forall i\, (2-\gamma)(1-c_i^e) > \gamma \sum_{j \neq i}(c_i^e - c_j^e)$, Nash equilibrium quantities and profits are equal to*

$$q_i^* = \frac{2 - \gamma - (2 + \gamma(m-2))c_i^e + \gamma \sum_{j \neq i} c_j^e}{(2-\gamma)(2+(m-1)\gamma)}, \ \Pi_i^e = (q_i^*)^2.$$

**Bertrand competition** Each firm acts by setting the price $p_i$ for their product, which determines the quantities (5) and expected profits (6). The following lemma describes the Nash equilibrium of this game.

**Lemma 4.4.** *Assume that the expected marginal costs are equal to $c_1^e, \ldots, c_m^e$, and companies maximize their profits (6) in the Bertrand competition game. If $\forall i \, d_1(1 - c_i^e) > d_3 \sum_{j \neq i}(c_i^e - c_j^e)$, Nash equilibrium prices and profits are equal to*

$$p_i^* = \frac{d_1 + d_2 c_i^e + d_3 \sum_{j \neq i} c_j^e}{d_4}, \; \Pi_i^e = \frac{(1 + \gamma(m+2))(p_i^* - c_i^e)^2}{(1 - \gamma)(1 + \gamma(m-1))},$$

*where $d_1, \ldots, d_4$ depend only on $\gamma$ and $m$.*

In both lemmas, the technical conditions $\forall i \, (2 - \gamma)(1 - c_i^e) > \gamma \sum_{j \neq i}(c_i^e - c_j^e)$ and $\forall i \, d_1(1 - c_i^e) > d_3 \sum_{j \neq i}(c_i^e - c_j^e)$ ensure that every firm produces a positive amount of good in the equilibrium. We prove these Lemmas in Sections A.2 and A.3.

## 4.2 Data impact model

In order to reason strategically about the impact of sharing data with others, firms need to understand how additional data impacts their costs. Here, we consider the case where all datasets are sampled from the same distribution $\mathcal{D}$. While heterogeneity is an important concern in collaborative learning, we focus on the homogeneous case since heterogeneity is orthogonal to the incentives arising from market equilibrium considerations.

Homogeneity allows us to consider the expected costs in form $c_i^e = c^e(n_i^c)$, where $n_i^c = |D_i^c|$ is the number of points the company $i$ has access to. Using the examples below, we motivate the following functional form for $c^e(n)$

$$c^e(n) = a + \frac{b}{n^\beta}, \; \beta \in (0, 1]. \tag{7}$$

In this setting, $\beta$ indicates the *difficulty of the learning task*, higher $\beta$ corresponds to a simpler task (Tsybakov, 2004), $a$ represents the marginal costs of production given a perfect machine learning model, while $a + b$ corresponds to the cost of production without machine learning optimizations. Additionally, we assume that $a < 1$ and $\frac{b}{1-a}$ is small enough, so that the technical requirements of Lemmas 4.1, 4.3, and 4.4 are satisfied (see Equations (10) and (11)). Intuitively, this requirement ensures that firms do not exit the market during the competition stage.

In the examples below, we consider a setting where a company needs to perform action $\boldsymbol{s} \in \mathbb{R}^n$ during production that will impact its costs. However, there is uncertainty in the production process coming from random noise $\boldsymbol{X}$ valued in $\mathbb{R}^m$. Thus, the cost of production is a random function $c(\boldsymbol{s}, \boldsymbol{X}) \colon \mathbb{R}^n \times \mathbb{R}^m \to \mathbb{R}_+$.

**Asymptotic normality** Assume that the firms use background knowledge about their production processes in the form of a structural causal model of $\boldsymbol{X}$. However, they do not know the exact parameters of the model and use data to estimate them. Suppose a firm uses the maximum likelihood estimator to find the parameters and chooses the optimal action $\boldsymbol{s}_{\text{fin}}$ based on this estimate. In this case, the result of optimization $\mathbf{E}_{\boldsymbol{X}}(c(\boldsymbol{s}_{\text{fin}}, \boldsymbol{X}))$ will approximately have a generalized chi-square distribution (Jones, 1983), resulting in the following approximation

$$\mathbf{E}_{D, \boldsymbol{X}}(c(\boldsymbol{s}_{\text{fin}}, \boldsymbol{X})) \approx a + \frac{b}{n}. \tag{8}$$

We refer to Appendix A.4 for further details. This result implies the same dependence of the expected marginal costs on the total number of samples $n$ as Equation (7) with $\beta = 1$.

**Stochastic optimization** A similar dependence arises when the company uses stochastic optimization to directly optimize the expected cost $c(\boldsymbol{s}) = \mathbf{E}_{\boldsymbol{X}}(c(\boldsymbol{s}, \boldsymbol{X}))$. If the company uses an algorithm with provable generalization guarantees (e.g., SGD with a single pass over the data, Bubeck 2015) and the function $c(\boldsymbol{s})$ is strongly convex, the outcome $\boldsymbol{s}_{\text{fin}}$ will satisfy

$$\mathbf{E}_D(c(\boldsymbol{s}_{\text{fin}}) - c(\boldsymbol{s}^*)) = \mathrm{O}\left(\frac{1}{n}\right),$$

where $\boldsymbol{s}^*$ is the optimal action, resulting in the same dependence as in the previous paragraph.

**Statistical learning theory**    Another justification for the dependency on the number of data points comes from statistical learning theory. The observations from this subsection are inspired by similar arguments in Karimireddy et al. (2022). Assume that a firm trains a classifier used in the production process. In this case, the cost $c$ depends on the classifier accuracy $a$, and the cost overhead will satisfy

$$c(a) - c(a^*) \approx -c'(a^*)(a^* - a),$$

where $a^*$ is the optimal accuracy achievable by a chosen family of classifiers. If the family of classifiers has a finite VC dimension $d$, a well-known statistical bound (Shalev-Shwartz & Ben-David, 2014) gives

$$a^* - a = \mathrm{O}\left(\sqrt{\frac{d}{n}}\right),$$

motivating the same dependence as Equation (7) with $\beta = 1/2$.

## 5    Data sharing between two firms

Having introduced example market and data impact models, we can specify a negotiation scheme and quantitatively analyze the data-sharing problem. Through this, we aim to obtain qualitative insights into how market parameters impact data-sharing decisions. Following conventional economic logic (Tirole, 1988; Carlton & Perloff, 1990), we start with the simplest possible collaboration scheme, in which two companies make a binary decision of whether to share all their data with each other. In the next sections, we proceed to the more complicated situations of partial data sharing and data sharing between many parties.

**Full data sharing between two firms**    According to our framework, both companies compare their expected profits at the Nash equilibrium (from Lemma 4.3 or 4.4) for two cases, when they collaborate and when they do not. Then *they share data with each other if and only if they both expect an increase in profits from this action.*

*Remark* 5.1.  Notice that we can incorporate some legal requirements into this scheme by redefining full data-sharing action. For example, if sharing consent constraints, araising from copyright or other data ownership constraint, are present, we could assume that the companies will share only a part of their dataset for the collaborative learning. While, previously, the competitor gets access to all data of the firm $n_{-i}^{\text{share}} = n_{-i} + n_i$, now they will get access to only data of people who agree with data sharing $n_{-i}^{\text{share}} = n_{-i} + n_{i,\text{consent}}$.

For Cournot competition, Lemma 4.3 and Equation (7) give the following collaboration criterion

$$\forall i \quad \Pi_{\text{share}}^e > \Pi_{i,\text{ind}}^e \iff (2-\gamma)(n_i^{-\beta} - (n_1 + n_2)^{-\beta}) > \gamma(n_{-i}^{-\beta} - n_i^{-\beta}),$$

where $n_{-i}$ is the size of the data of the player that is not $i$, $\Pi_{i,\text{share}}^e$ is the expected profit in collaboration, and $\Pi_{i,\text{ind}}^e$ is the expected profit when no collaboration occurs.

Similarly, in the case of Berntrand competition, Lemma 4.4 gives

$$\forall i \quad \Pi_{\text{share}}^e > \Pi_{i,\text{ind}}^e \iff (2-\gamma-\gamma^2)(n_i^{-\beta} - (n_1 + n_2)^{-\beta}) > \gamma(n_{-i}^{-\beta} - n_i^{-\beta}).$$

The theorem below describes the properties of these criteria (see proof in Appendix A.5).

**Theorem 5.2.** *In the case of $\gamma \leq 0$, it is profitable for the firms to collaborate. In the case of $\gamma > 0$, there exists a value $x_t(\gamma, \beta)$, where $t \in \{\text{Bertrand}, \text{Cournot}\}$ is the type of competition, such that, for the firm $i$, it is profitable to collaborate only with a competitor with enough data:*

$$\Pi_{\text{share}}^e > \Pi_{i,\text{ind}}^e \iff \frac{n_{-i}}{n_1 + n_2} > x_t(\gamma, \beta).$$

*The function $x_t$ has the following properties:*

1. *$x_t(\gamma, \beta)$ is increasing in $\gamma$.*

2. *$x_{\text{Bertrand}} \geq x_{\text{Cournot}}$.*

3. *$x_t(\gamma, \beta)$ is increasing in $\beta$.*

**Discussion** The theorem above indicates that the *firms are more likely to collaborate when either the market is less competitive* (properties 1 and 2) *or the learning task is harder* (property 3). Indeed, the threshold $x$ becomes smaller when the products are less similar ($\gamma$ is bigger), making the market less competitive. Similarly, $x$ becomes smaller in the Cournot setting since it is known to be less competitive than the Bertrand one (Shapiro, 1989). Finally, when the learning task is harder ($\beta$ is smaller), $x$ decreases, making collaboration more likely. Intuitively, it happens because the decrease in cost (7) from a single additional data point is higher for smaller $\beta$.

In the supplementary material, we explore several extensions for the two firms case. In Appendix C.1 and Appendix C.2, we look at different cost and utility functions, respectively. In Appendix C.3, we consider the case where $a$, $b$ and/or $\beta$ might differ among firms and derive an analog of Theorem 5.2. In Appendix C.4, we consider the data-sharing problem with two companies with heterogeneous data in the context of mean estimation. In Appendix E, we look at the welfare implications of data sharing.

# 6 Partial data sharing

In this section, we study a two-companies model in which companies can share any fraction of their data with competitors. For brevity, we only study this model in the case of Cournot competition. In this setup, each firm $F_i$ chooses to share a fraction $\lambda_i \in [0, 1]$ of data with its competitor and expects its costs to be

$$c_i^e = c^e(n_i + \lambda_{-i} n_{-i}) = a + b(n_i + \lambda_{-i} n_{-i})^{-\beta},$$

where $\lambda_{-i}$ is the fraction of data shared by the company that is not $i$.

*Remark* 6.1. Notice that we can incorporate some legal requirements into this scheme by constraining the action spaces of participants. For example, one can implement sharing consent constraints, araising from copyright or other data ownership constraint, by constraining the firms' choices of $\lambda$. While, previously, the firms were able to share all data $\lambda \in [0, 1]$, now they can share only data of people who agree with data sharing $\lambda \in [0, \lambda_{\text{consent}}]$.

We will use the Nash bargaining solution (Binmore et al., 1986) to describe the bargaining process between the firms. Namely, the firms will try to compromise between themselves by maximizing the following Nash product

$$\max_{\lambda_i \in [0,1]} \left( \Pi_1^e(\lambda_1, \lambda_2) - \Pi_1^e(0,0) \right) \left( \Pi_2^e(\lambda_1, \lambda_2) - \Pi_2^e(0,0) \right) \text{ s.t. } \forall i \; \Pi_i^e(\lambda_1, \lambda_2) - \Pi_i^e(0,0) \geq 0. \quad (9)$$

The Nash bargaining solution is a commonly used model for two-player bargaining, as it results in the only possible compromise satisfying invariance to the affine transformations of profits, Pareto optimality, independence of irrelevant alternatives, and symmetry (Binmore et al., 1986). The following theorem describes the firms' behavior in this setup (see proof in Appendix A.6).

**Theorem 6.2.** *W.l.o.g. assume that $n_1 \geq n_2$. Additionally, assume $(1 - a) \gg b$ (in particular, $b < 5(1 - a)$). The solution $(\lambda_1^*, \lambda_2^*)$ to the problem (9) is*

$$\lambda_1^* = \tilde{\lambda}_1 + \mathrm{O}\left( \frac{b}{1-a} \right), \; \lambda_2^* = 1, \text{ where}$$

$$\tilde{\lambda}_1 = \begin{cases} \frac{n_2}{n_1} \left( \left( 1 - \frac{4+\gamma^2}{4\gamma} \left( \frac{n_2}{n_1} \right)^\beta \left( 1 - \left( \frac{n_1}{n_1+n_2} \right)^\beta \right) \right)^{-1/\beta} - 1 \right), & \frac{4+\gamma^2}{n_1^\beta} \leq \frac{4\gamma}{n_2^\beta} + \frac{(2-\gamma)^2}{(n_1+n_2)^\beta}, \\ 1 & otherwise. \end{cases}$$

*Moreover, $\tilde{\lambda}_1$ is a decreasing function in $\gamma$ and $\beta$, and $\tilde{\lambda}_1 = \frac{4+\gamma^2}{4\gamma} \left( \frac{n_2}{n_1} \right)^{\beta+2} (1 + \mathrm{o}(1))$ when $n_2/n_1 \to 0$.*

**Discussion** We can see that the results of Theorem 6.2 are similar to those in Section 5. When there is a big difference between the amount of data ($n_1 \gg n_2$), the big firm does not collaborate with the small firm ($\lambda_1^* = \mathrm{O}((n_2/n_1)^{\beta+2})$). However, when the firms have similar amounts of data, the large firm will share almost all of it ($\lambda_1^* \approx 1$), and the sharing proportion increases with a decrease in competition (smaller $\gamma$) and an increase in learning task hardness (smaller $\beta$). In contrast to the previous results, the smaller firm always shares all its data with the competitor ($\lambda_2^* = 1$).

# 7 Data sharing between many firms

In this section, we investigate the data-sharing problem for many companies. For brevity, we only consider the case of Cournot competition. The key challenge in the case of many companies is the variety of possible coalitional structures. Moreover, since the companies may have conflicting incentives and data sharing requires mutual agreement, it is not immediately clear how to assess the "plausibility" of a particular structure.

We discuss one possible way to solve this problem using non-cooperative game theory (Kóczy, 2018). We assume a coalition formation process between the firms, model it as a non-cooperative game, and study its Nash equilibria. The non-cooperative approach often offers a unique *sub-game perfect equilibrium*, a standard solution notion in sequential games (Mas-Colell et al., 1995). However, this equilibrium depends on the assumptions about the bargaining process. For this reason, in Appendix D we also study alternative formulations: a cooperative setting and two non-cooperative settings with only one non-singleton coalition.

**Non-cooperative data-sharing game**   We order the firms in a decreasing data size order so that firm $F_1$ has the largest dataset. In this order, the firms propose forming a coalition with their competitors. First, the largest company makes up to $2^{m-1}$ proposals. For any offer, each invited firm accepts or declines it in the decreasing data size order. If anyone disagrees, the coalition is rejected, and the first firm makes another offer. Otherwise, the coalition is formed, and all companies in it leave the game. After the first company forms a coalition or exhausts all of its offers, the second firm, in the decreasing data size order, begins to propose, etc.

We use the standard backward induction procedure (Mas-Colell et al., 1995) to calculate the sub-game perfect equilibrium of this game. This method solves the game from the end to the beginning. First, the algorithm looks at all possible states one step before the end of the game. Since the actions of the last player are rational, the algorithm can identify these actions accurately. This way, the algorithm moves one step earlier in the coalition formation process and can now identify the action taken before this state occurred. The procedure iterates until the start of the game is reached.

**Experiments**   We use the procedure described above to empirically test the conclusions of the previous sections. We sample $m$ dataset sizes, one for each firm, from a distribution $P = \mathrm{N}(\mu, \sigma^2)$ clipped at 1 form below. Then, we calculate the equilibrium of the resulting data-sharing game and

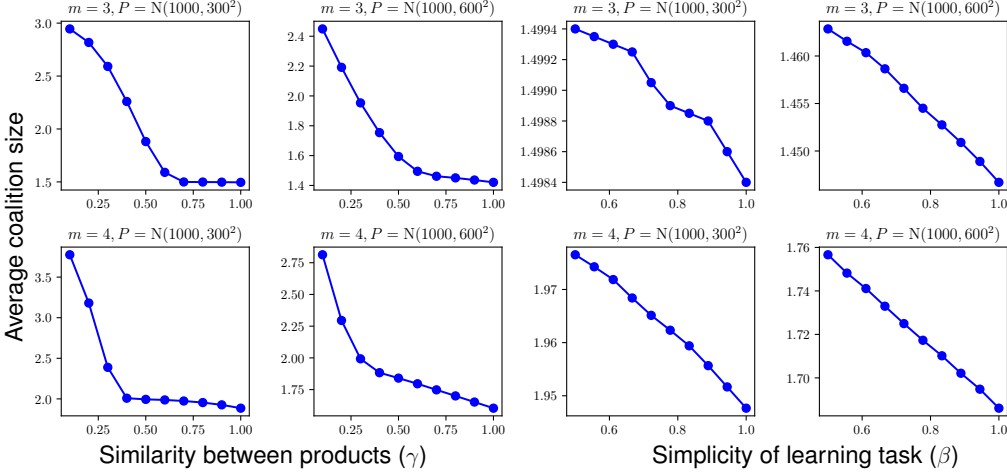

Figure 1: The dependence of the average coalition size on $\gamma$ (left part) and $\beta$ (right part) in the synthetic experiments. The $y$-axes report the mean of the average size of the coalitions in the equilibrium partition, where mean is taken over 10000 Monte Carlo simulations of the game. See the main text for details.

compute the average size (number of companies) of the resulting coalitions at the equilibrium, as a measure of the extend to which companies collaborate.

We repeat the experiment 10000 times, for fixed values of $m, \gamma, \beta, \mu, \sigma$ and compute the mean of the average coalition size over these runs. Our simulation solves each instance of the data-sharing game exactly and average it over a big number of independent runs, which makes our results very precise.

Repeating the experiment for different values of $\gamma$ and $\beta$, we can observe the dependence of this average coalition size on the two parameters of interest. In Figure 1 we plot these dependencies, for different values of $m \in \{3, 4\}$ and $\sigma \in \{300, 600\}$. When varying one of these parameters, the default values for the other one is $\gamma = 0.8$ and $\beta = 0.9$. We also provide results for other values of $m$ and $\sigma$ in Appendix F.

As we can see, the conclusion of Theorem 5.2 transfer to this experiment: the average size of the coalitions and thus cooperation incentives, are decreasing in both $\gamma$ and $\beta$, across all experimental setups.

## 8    Conclusion

In this work, we introduced and studied a framework for data-sharing among market competitors. The framework is modular and consists of three components—market model, data impact, and collaboration scheme—which can be specified for any particular application. To examine the effects of various parameters, we instantiated the framework using a conventional market model and a natural model of data impact grounded in learning theory. We then studied several data-sharing games within these models.

Our findings indicate that the characteristics of competition may have a significant effect on the data-sharing decisions. Specifically, we found that higher product differentiation generally increases the willingness for collaboration, as does the complexity of the learning task. Interestingly, our results suggests that the firms might not want to participate in the federated learning even in the absence of privacy concerns. On the other hand, we also predict that data-sharing collaborations might emerge between competitors even without any external regulations. We hope our study will inspire further in-depth investigations into the nuanced trade-offs in data sharing, allowing competition and collaboration to coexist in data-driven environments.

**Limitations and future work**    While our general framework can describe many data-sharing settings, the quantitative results rely on several important assumptions. First, we assume that the firms' data are homogeneous. We expect that designing a suitable model in the case of significant heterogeneity is a significant challenge orthogonal to the focus of our work (e.g., Gulrajani & Lopez-Paz, 2021). Moreover, in some cases, additional data from a different distribution may damage model performance, yielding orthogonal data-sharing considerations (Donahue & Kleinberg, 2021).

Second, we assume that legal constraints are not present or are automatically fulfilled in all our settings. However, we see the evaluation of AI regulatory frameworks as an important direction for future work. The same applies to training cost considerations.

Finally, we do not consider sociological aspects, such as reputational effects (e.g., reputation loss due to a refusal to share data or a poorly communicated decision to share data). Due to the inherent non-rivalry of data (Karimireddy et al., 2022), sociological modeling, e.g., the theory of collective action, may provide further valuable insights into the data-sharing problem.

We see our contributions as mostly conceptual and do not aim to provide a fully realistic model that can directly inform practitioners about the benefits of data-sharing decisions. However, we hope our results will remain qualitatively valid in real-world settings since we used a market model widely adopted in the economic theoretical literature (Choné & Linnemer, 2019) and a data impact model motivated by established theoretical frameworks in machine learning (Tsybakov, 2004). Also, we hope that the three-component decomposition of the data-sharing problem will help practitioners leverage their situation-specific knowledge of their market and ML models to build more accurate models for their specific needs.

## Acknowledgments

This research was partially funded by the Ministry of Education and Science of Bulgaria (support for INSAIT, part of the Bulgarian National Roadmap for Research Infrastructure). The authors would like to thank Florian Dorner, Mark Vero, Nikola Jovanovic and Sam Motamed for providing helpful feedback on earlier versions of this work.

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

# Supplementary Material

The supplementary material is structured as follows.

- Appendix A contains the proofs of all results in the main text.
- Appendix B contains further examples of the data-sharing problem and costs.
- Appendix C contains several extensions of the setting in Section 5.
- Appendix D contains alternative models of coalition formation between many firms.
- Appendix E contains the welfare analysis of the data-sharing problem between two firms.
- Appendix F presents additional simulations for Section 7.

## A Proofs

### A.1 Proof of Lemma 4.1

First, we will solve the same consumption problem without positivity constraints

$$\max_{g, q_1, \ldots, q_m} u(g, q_1, \ldots, q_m) \text{ s.t. } g + \sum_{i=1}^{m} p_i q_i \leq B.$$

Using the KKT conditions for $q_i$ we get

$$0 = \frac{\partial L}{\partial q_i} = \frac{\partial u}{\partial q_i} - \lambda p_i,$$

where $\lambda$ is the Lagrange multiplier for the budget constraint. The first-order condition for $g$ gives

$$0 = 1 - \lambda \implies \lambda = 1.$$

Therefore, the first-order conditions for $q_1, \ldots, q_m$ look like

$$0 = 1 - q_i - \sum_{j \neq i} q_j - p_i \implies p_i = 1 - q_i - \gamma \sum_{j \neq i} q_j.$$

Notice that

$$\sum_i p_i = m - \sum_i q_i - \gamma(m-1) \sum_i q_i \implies \sum_i q_i = \frac{m - \sum_i p_i}{1 + \gamma(m-1)}.$$

Therefore,

$$q_i = \frac{(1 + \gamma(m-1))(1 - p_i) - \gamma(m - \sum_i p_i)}{(1 - \gamma)(1 + \gamma(m-1))} = \frac{1 - \gamma - (1 + \gamma(m-2))p_i - \gamma \sum_{j \neq i} p_j}{(1 - \gamma)(1 + \gamma(m-1))}.$$

This solution is a global maximizer because $u(g, q_1, \ldots, q_m)$ is strongly concave in $q_1, \ldots, q_m$ and linear in $g$.

Since the auxiliary problem has less constraints than the original problem, a solution to the auxiliary problem will be a solution to the original problem as long as all quantities are non-negative. These constraints give the following restrictions on $p_1, \ldots, p_m$ and $B$

$$\forall i \ (1 - \gamma)(1 - p_i) \geq \gamma \sum_{j \neq i} (p_j - p_i),$$

$$B \geq \sum_{i=1}^{m} q_i^* p_i.$$

The first series of inequalities hold when prices are less than one and close enough. In our setting, it is achieved when $1 > a$ and $(1 - a) \gg b$, where $a$ and $b$ are parameters from Equation (7) (since Equations (10) and (11) hold). The last equation holds when $B$ is big enough, for example, when $B > m$.

## A.2 Proof of Lemma 4.3

The profit maximization tasks of the firms are

$$\max_{q_i} \mathbf{E}((p_i - c_i)q_i) = \mathbf{E}(1 - \gamma \sum_{j \neq i} q_j - c_i)q_i - q_i^2.$$

Using the first-order conditions, we get

$$q_i^*(q_{-i}) = \frac{\mathbf{E}(1 - c_i - \gamma \sum_{j \neq i} q_j)}{2}.$$

Both equilibrium quantities should be the best responses to the opponent's expected amounts

$$q_i^* = \frac{\mathbf{E}(1 - c_i - \gamma \sum_{j \neq i} q_j^*)}{2}.$$

Therefore,

$$\sum_i q_i^* = \frac{\mathbf{E}(m - \sum_i c_i - (m-1)\gamma \sum_i q_i^*)}{2},$$

Notice that

$$\mathbf{E}(q_i^*) = \frac{\mathbf{E}\,\mathbf{E}(\dots)}{2} = \frac{\mathbf{E}(\dots)}{2} = q_i^*.$$

So,

$$\sum_i q_i^* = \frac{m - \sum_i c_i^e}{2 + \gamma(m-1)}.$$

The best response equation gives

$$q_i^* = \frac{1 - c_i^e - \gamma \sum_j q_j^* + \gamma q_i^*}{2} \implies q_i^* = \frac{2 - \gamma - (2 + \gamma(m-2))c_i^e + \gamma \sum_{j \neq i} c_j^e}{(2 - \gamma)(2 + (m-1)\gamma)}.$$

We get some expression for the quantities produced by the firms. However, we need to ensure that these quantities are positive. This property is equivalent to the following inequalities

$$\forall i \ (2 - \gamma)(1 - c_i^e) \geq \gamma \sum_{j \neq i} (c_i^e - c_j^e). \tag{10}$$

These inequalities hold, for example, when $1 > a$ and $(1 - a) \gg b$, where $a$ and $b$ are parameters from Equation (7).

Now, notice that the expected profit function is a second-degree polynomial in $q_i$

$$\Pi_i^e = -q_i^2 + \beta q_i.$$

Since the maximum value is achieved at $q_i^*(q_{-i})$, this polynomial looks like

$$\Pi_i^e = -(q_i - q_i^*(q_{-i}))^2 + q_i^*(q_{-i})^2.$$

And it gives the following equilibrium expected profits

$$\Pi_i^e = (q_i^*)^2.$$

## A.3 Proof of Lemma 4.4

The expected profit maximization tasks of the firms are

$$\max_{p_i} \mathbf{E}\left( (p_i - c_i)\left( \frac{1 - \gamma - (1 + \gamma(m-2))p_i + \gamma \sum_{j \neq i} p_j}{(1 - \gamma)(1 + \gamma(m-1))} \right) \right).$$

Thus, the best responses are

$$p_i^*(p_{-i}) = \frac{\mathbf{E}(1 - \gamma + (1 + \gamma(m-2))c_i + \gamma \sum_{j \neq i} p_j)}{2(1 + \gamma(m-2))}.$$

Notice that
$$\mathbf{E}(p_i^*) = \frac{\mathbf{E}\,\mathbf{E}(\dots)}{2(1 + \gamma(m-2))} = \frac{\mathbf{E}(\dots)}{2(1 + \gamma(m-2))} = p_i^*.$$

Thus,
$$p_i^* = \frac{1 - \gamma + (1 + \gamma(m-2))c_i^e + \gamma \sum_{j \neq i} p_j^*}{2(1 + \gamma(m-2))}.$$

If we sum these equations, we get
$$\sum_i p_i^* = \frac{(1 - \gamma)m + (1 + \gamma(m-2)) \sum_i c_i^e + \gamma(m-1) \sum_i p_i^*}{2(1 + \gamma(m-2))} \implies$$
$$\sum_i p_i^* = \frac{(1 - \gamma)m + (1 + \gamma(m-2)) \sum_i c_i^e}{2 + \gamma(m-3)}.$$

After substitution of this sum into the best response equation, we have
$$p_i^* = \frac{d_1 + d_2 c_i^e + d_3 \sum_{j \neq i} c_j^e}{d_4},$$
$$d_1 = 2 + \gamma(2m - 5) - \gamma^2(2m - 3),$$
$$d_2 = 2 + 3\gamma(m-2) + \gamma^2(m^2 - 4m + 4),$$
$$d_3 = \gamma + \gamma^2(m-2),$$
$$d_4 = 4 + 6\gamma(m-2) + \gamma^2(2m^2 - 9m + 9).$$

Here, we again need to ensure that quantities produced by the firms are positive. This property is equivalent the following inequalities
$$\forall p_i - c_i^e \geq 0 \iff d_1(1 - c_i^e) \geq d_3 \sum_{j \neq i} (c_i^e - c_j^e). \tag{11}$$

These inequalities hold, for example, when $1 > a$ and $(1 - a) \gg b$, where $a$ and $b$ are parameters from Equation (7).

Notice that the expected profit of the firm is a second degree polynomial
$$\Pi_i^e = -\alpha p_i^2 + 2\beta p_i - \gamma.$$

Since best response $p_i^*(p_{-i})$ is a maximizer of this polynomial, we get
$$\Pi_i^e = -\alpha(p_i - p_i^*(p_{-i}))^2 + \delta.$$

Also, notice that
$$\Pi_i^e(c_i^e, p_{-i}) = \mathbf{E}((c_i^e - c_i)q_i(c_i^e, p_{-i})) = 0.$$

Therefore,
$$\Pi_i^e = \alpha((c_i^e - p_i^*(p_{-i}))^2 - (p_i - p_i^*(p_{-i}))^2).$$

Direct calculations show that
$$\alpha = \frac{1 + \gamma(m+2)}{(1 - \gamma)(1 + \gamma(m-1))}.$$

These equations imply the following equilibrium profits
$$\Pi_i^e(p_i^*, p_{-i}^*) = \frac{(1 + \gamma(m+2))(p_i^* - c_i^e)^2}{(1 - \gamma)(1 + \gamma(m-1))},$$

## A.4  Derivation of Equation (8)

Formally, we assume that the cost of production of one unit of good $C$ is a random variable that depends on random noise $\boldsymbol{X} \in \mathbb{R}^m$ and the decisions of a firm $\boldsymbol{s} \in \mathbb{R}^n$. The firm knows that the noise is distributed according to a density function
$$\boldsymbol{X} \sim f(\cdot, \boldsymbol{\theta}), \boldsymbol{\theta} \in \mathbb{R}^d.$$

However, it does not know the parameters of the distribution $\boldsymbol{\theta}$.

The firm wants to minimize its marginal cost on average

$$c(\boldsymbol{s}) = \mathbf{E}_{\boldsymbol{X} \sim f(\cdot, \boldsymbol{\theta})}(C(\boldsymbol{s}, \boldsymbol{X})).$$

To achieve it, the firm decides to use the maximum likelihood estimates $\widehat{\boldsymbol{\theta}}$ of $\boldsymbol{\theta}$ using its data about the realizations of the noise. After that, the firm chooses an action that minimizes an estimated cost on average

$$\boldsymbol{s}_{\text{fin}} = \arg \min_{\boldsymbol{s}} \mathbf{E}_{\boldsymbol{X} \sim f(\cdot, \widehat{\boldsymbol{\theta}})}(C(\boldsymbol{s}, \boldsymbol{X})).$$

The resulting cost $c(\boldsymbol{s}_{\text{fin}})$ is a random variable because of randomness of the firm's sample $S$. However, the asymptotic normality of MLE allows us to reason about a cost distribution.

We use the delta method to achieve this goal. By the asymptotic normality,

$$\sqrt{n}(\widehat{\boldsymbol{\theta}} - \boldsymbol{\theta}) \xrightarrow{d} \mathrm{N}(0, \mathbf{I}(\boldsymbol{\theta})^{-1}).$$

Further, by the delta method,

$$\sqrt{n}(\boldsymbol{s}_{\text{fin}} - \boldsymbol{s}^*) \xrightarrow{d} \sqrt{n} \left( \frac{\partial \boldsymbol{s}}{\partial \boldsymbol{\theta}} \right)^{\mathsf{T}} (\widehat{\boldsymbol{\theta}} - \boldsymbol{\theta}).$$

Finally, by the second-order delta method,

$$n(c(\boldsymbol{s}_{\text{fin}}) - c(\boldsymbol{s}^*)) \xrightarrow{d} n \nabla c(\boldsymbol{s}^*)^{\mathsf{T}} (\boldsymbol{s}_{\text{fin}} - \boldsymbol{s}^*) + n(\boldsymbol{s}_{\text{fin}} - \boldsymbol{s}^*)^{\mathsf{T}} \frac{\nabla^2 c(\boldsymbol{s}^*)}{2}(\boldsymbol{s}_{\text{fin}} - \boldsymbol{s}^*)$$

$$= n(\boldsymbol{s}_{\text{fin}} - \boldsymbol{s}^*)^{\mathsf{T}} \frac{\nabla^2 c(\boldsymbol{s}^*)}{2}(\boldsymbol{s}_{\text{fin}} - \boldsymbol{s}^*).$$

Thus, the marginal cost will be distributed approximately as generalized chi-squared distribution. The expected marginal cost will be approximately

$$c^e = \mathbf{E}_S(c(\boldsymbol{s}_{\text{fin}})) \approx c(\boldsymbol{s}^*) + \frac{1}{2n} \operatorname{Tr} \left( \nabla^2 c(\boldsymbol{s}^*) \left( \frac{\partial \boldsymbol{s}}{\partial \boldsymbol{\theta}} \right)^{\mathsf{T}} \mathbf{I}(\boldsymbol{\theta})^{-1} \frac{\partial \boldsymbol{s}}{\partial \boldsymbol{\theta}} \right).$$

Therefore, one of the natural choices for the dependency of expected marginal cost on the number of data points is

$$c^e(n) = a + \frac{b}{n}.$$

## A.5  Proof of Theorem 5.2

In the case $\gamma \le 0$, the collaboration is always profitable because the right-hand side of the criterion is less than zero, and the left-hand side of the criterion is not less than zero.

In the case $\gamma > 0$, the criteria for the firm $i$ are the following. In the case of Cournot competition, we have

$$(2 - \gamma)(n_i^{-\beta} - (n_1 + n_2)^{-\beta}) > \gamma(n_{-i}^{-\beta} - n_i^{-\beta})$$
$$\iff f_{\text{Cournot}}(x, \gamma, \beta) := 2x^\beta - \gamma(1 - x)^\beta - (2 - \gamma)x^\beta(1 - x)^\beta > 0,$$

where $x = n_{-i}/(n_1 + n_2)$. The right-hand side of the criterion $f$ is increasing in $x$, $f_{\text{Cournot}}(0, \gamma, \beta) < 0$, and $f_{\text{Cournot}}(1, \gamma, \beta) > 0$. Thus, there exists a break-even point $x_{\text{Cournot}}(\gamma, \beta)$ such that

$$f_{\text{Cournot}}(x, \gamma, \beta) \begin{cases} < 0, & x < x_{\text{Cournot}}(\gamma, \beta), \\ = 0, & x = x_{\text{Cournot}}(\gamma, \beta), \\ > 0, & x > x_{\text{Cournot}}(\gamma, \beta). \end{cases}$$

Similarly, a break-even point exists in the Bertrand case.

Now, we prove the properties of $x$.

The first property follows from the fact that $f$ is decreasing in $\gamma$. The second property follows from the fact that $f_{\text{Cournot}} > f_{\text{Bertrand}}$. The proof of the third property is more complicated.

In the case of Cournot, the proof is the following. Let $x(\beta) = x_{\text{Cournot}}(\gamma, \beta)$. We want to show that $x(\beta)$ is increasing in $\beta$. To achieve it, we will show that $f_{\text{Cournot}}(x(\beta), \gamma, \beta + \epsilon) < 0$. From the definition,

$$x(\beta) = \left( \frac{\gamma(1 - x(\beta))^\beta}{2 - (2 - \gamma)(1 - x(\beta))^\beta} \right)^{\frac{1}{\beta}}.$$

So,

$$f_{\text{Cournot}}(x(\beta), \gamma, \beta + \epsilon) < 0$$

$$\iff \left( \frac{\gamma(1 - x(\beta))^\beta}{2 - (2 - \gamma)(1 - x(\beta))^\beta} \right)^{\frac{1}{\beta}} < \left( \frac{\gamma(1 - x(\beta))^{\beta+\epsilon}}{2 - (2 - \gamma)(1 - x(\beta))^{\beta+\epsilon}} \right)^{\frac{1}{\beta+\epsilon}}$$

$$\iff \frac{\gamma}{2} \left( \frac{2 - (2 - \gamma)(1 - x(\beta))^{\beta+\epsilon}}{\gamma} \right)^{\frac{\beta}{\beta+\epsilon}} + \frac{2 - \gamma}{2}(1 - x)^\beta < 1.$$

The last inequality follows from the concavity of the function $x^{\frac{\beta}{\beta+\epsilon}}$. The proof is similar in the Bertrand case.

### A.6  Proof of Theorem 6.2

Direct computations show

$$\Pi_i^e(\lambda_1, \lambda_2) - \Pi_i^e(0,0) =$$
$$(2c_i^e(n_i) - \gamma c_{-i}^e(n_{-i}) - 2c_i^e(n_i + n_{-i}\lambda_{-i}) + \gamma c_{-i}^e(n_{-i} + n_1\lambda_i)) \times$$
$$\frac{(4 - 2\gamma - 2c_i^e(n_i) + \gamma c_{-i}^e(n_{-i}) - 2c_i^e(n_i + n_{-i}\lambda_{-i}) + \gamma c_{-i}^e(n_{-i} + n_i\lambda_i))}{(4 - \gamma^2)^2}.$$

Denote

$$u_i = \frac{1}{n_i^\beta} - \frac{1}{(n_i + \lambda_{-i}n_{-i})^\beta}.$$

It gives

$$c_i^e(n_i) - c_i^e(n_i + \lambda_{-i}n_{-i}) = a + \frac{b}{n_i^\beta} - a - \frac{b}{(n_i + n_{-i}\lambda_{-i})^\beta} = bu_i.$$

So, the profit gain will be equal to

$$\Pi_i^e(\lambda_1, \lambda_2) - \Pi_i^e(0,0) = (2u_i - \gamma u_{-i}) \left( 1 - \frac{4\xi}{n_i^\beta} + \frac{2\gamma\xi}{n_{-i}^\beta} + \xi(2u_i - \gamma u_{-i}) \right) \frac{b(4 - 2\gamma)(1 - a)}{(4 - \gamma^2)^2},$$

where $\xi = \frac{b}{(4 - 2\gamma)(1 - a)}$. Denote

$$h_i = \frac{1}{n_i^\beta} - \frac{1}{(n_1 + n_2)^\beta},$$
$$g_i = \frac{4}{n_i^\beta} - \frac{2\gamma}{n_{-i}^\beta}.$$

Then Equation (9) will simplify to

$$\max_{u_i \in [0, h_i]} (2u_1 - \gamma u_2)(2u_2 - \gamma u_1)(1 - \xi g_1 + \xi(2u_1 - \gamma u_2))(1 - \xi g_2 + \xi(2u_2 - \gamma u_1)) \text{ s.t.} \forall i \ 2u_i - \gamma u_{-i} \geq 0.$$

Now, we will change variables

$$v_i := 2u_i - \gamma u_{-i} \implies u_i = \frac{2v_i + \gamma v_{-i}}{4 - \gamma^2},$$

which will give the following problem

$$\max_{v_i \geq 0} v_1 v_2 (1 - \xi g_1 + \xi v_1)(1 - \xi g_2 + \xi v_2) \text{ s.t. } \forall i \ 2v_i + \gamma v_{-i} \leq (4 - \gamma^2)h_i.$$

Notice that the restrictions $v_i \geq 0$ are not binding since the point $(v_1, v_2) = (\epsilon, \epsilon)$ delivers positive Nash product and satisfies the restrictions. However, one of the restrictions $2v_i + \gamma v_{-i} \leq (4 - \gamma^2)h_i$ should be binding because any internal point $(v_1, v_2)$ could be transformed to the point $((1+\epsilon)v_1, (1+\epsilon)v_2)$, which would increase the objective function and satisfy the restrictions.

First, consider the case $2v_1^* + \gamma v_2^* < (4 - \gamma^2)h_1$ at an optimum $(v_1^*, v_2^*)$. According to the previous paragraph, we have $2v_2^* + \gamma v_1^* = (4 - \gamma^2)h_2$, which gives $v_2^* - v_1^* > (2 + \gamma)(h_2 - h_1) \geq 0$. The derivative of the objective along the direction $(2, -\gamma)$ equals to

$$(2v_2^* - \gamma v_1^*)(1 - \xi g_1 + \xi v_1^*)(1 - \xi g_2 + \xi v_2^*) + \xi v_1^* v_2^* (2 - \gamma - \xi(2g_2 - \gamma g_1 - 2v_2^* + \gamma v_1^*)).$$

Since $2v_2^* - \gamma v_1^* > 0$, $|2g_2 - \gamma g_1 - 2v_2^* + \gamma v_1^*| \leq 10$, and $\xi \leq \frac{b}{2(1-a)} \leq \frac{1}{10}$, the derivative is positive. Thus, the point $(v_1^*, v_2^*)$ can not be optimal because a small shift in the direction $(2, -\gamma)$ would increase the objective and would not violate the restrictions. Therefore, in the optimum, we will have $2v_1^* = (4 - \gamma^2)h_1 - \gamma v_2^*$.

If we substitute the last expression into the objective function, the objective will become a fourth-degree polynomial of $v_2$. If the stationary point of the objective function $v_2^s$ satisfy the restriction $2v_2^s + \gamma v_1(v_2^s) \leq (4 - \gamma^2)h_2$, it will be a solution to the original problem. Otherwise, the solution will come form the equation $2v_2^0 + \gamma v_1(v_2^0) = (4 - \gamma^2)h_2$ since all other restrictions are not binding. So, the solution will be equal to $v_2^* = \min(v_2^s, v_2^0)$ (the restriction $2v_2 + \gamma v_1(v_2) \leq (4 - \gamma^2)h_2$ does not hold only when $v_2$ is big).

Consider the following relaxed problem

$$\max_{v_i \geq 0} v_1 v_2 (1 - \xi g_1 + \xi v_1)(1 - \xi g_2 + \xi v_2) \text{ s.t. } 2v_1 + \gamma v_2 = (4 - \gamma^2)h_1.$$

The stationary point will be the root of the derivative and hence will satisfy the following equation

$$(\gamma v_2^s - 2v_1(v_2^s))(1 - \xi g_1 + \xi v_1(v_2^s))(1 - \xi g_2 + \xi v_2^s) =$$
$$\xi v_1(v_2^s) v_2^s (2(1 - \xi g_1 + \xi v_1(v_2^s)) - \gamma(1 - \xi g_2 + \xi v_2^s))$$
$$\implies \gamma v_2^s - 2v_1(v_2^s) = \xi \frac{v_1(v_2^s) v_2^s (2(1 - \xi g_1 + \xi v_1(v_2^s)) - \gamma(1 - \xi g_2 + \xi v_2^s))}{(1 - \xi g_1 + \xi v_1(v_2^s))(1 - \xi g_2 + \xi v_2^s)}.$$

Since $-2 \leq g_i \leq 4$, $0 \leq v_i \leq 2$, and $v_1(v_2)v_2 \leq \frac{(4-\gamma^2)^2 h_1^2}{8\gamma}$, we get

$$|\gamma v_2^s - 2v_1(v_2^s)| \leq \xi \frac{(4 - \gamma^2)h_1^2}{8\gamma} \frac{2 - \gamma + 12\xi}{(1 - 4\xi)^2} = O(\xi).$$

So, the stationary point is approximately $v_2^s = \frac{(4-\gamma^2)h_1}{2\gamma} + O(\xi)$.

Therefore, the solution to the original problem is approximately

$$v_2^* = \min\left(\frac{(4 - \gamma^2)h_1}{2\gamma} + O(\xi), 2h_2 - \gamma h_1\right) = \min\left(\frac{(4 - \gamma^2)h_1}{2\gamma}, 2h_2 - \gamma h_1\right) + O(\xi).$$

Substituting everything back, we get the solution presented in the theorem statement.

Now, we will prove that $\tilde{\lambda}_1$ is non-increasing in $\gamma$.

First, the criterion function for choosing non-unit solution

$$\frac{(2 - \gamma)^2}{(n_1 + n_2)^\beta} + \frac{4\gamma}{n_2^\beta} - \frac{4 + \gamma^2}{n_1^\beta} = 4\gamma h_2 - (4 + \gamma^2)h_1.$$

is increasing in $\gamma$ since its derivative $4h_2 - 2\gamma h_1 \geq 0$ is positive. So, when $\gamma$ increases non-unit solution becomes more probable.

Second, the non-unit solution is decreasing in $\gamma$ since the function $\frac{4+\gamma^2}{\gamma} = \frac{4}{\gamma} + \gamma$ is decreasing in $\gamma$ on $(0, 1)$. (Note that $\tilde{\lambda}_1 = 1$, when $\gamma \leq 0$.)

Now, we will prove that $\tilde{\lambda}_1$ is non-increasing in $\beta$.

First, notice that the non-unit criterion is equivalent to

$$f(x, \beta) = 4\gamma(1-x)^\beta(1-x^\beta) - (4+\gamma^2)x^\beta(1-(1-x)^\beta) \geq 0,$$

where $x = \frac{n_2}{n_1+n_2}$. The criterion function $f$ is decreasing in $x$ and have one root $x(\beta)$ on the interval $[0, 1]$. We will prove that the root is shifting to the right, making more size pairs produce non-unit solution. Notice the criterion can be also expressed as

$$x \leq \left( \frac{4\gamma(1-x)^\beta}{4+\gamma^2 - (2-\gamma)^2(1-x)^\beta} \right)^{\frac{1}{\beta}}.$$

So, to prove that $x(\beta) \leq x(\beta + \epsilon)$ it is sufficient to show that

$$\left( \frac{4\gamma(1-x(\beta))^\beta}{4+\gamma^2 - (2-\gamma)^2(1-x(\beta))^\beta} \right)^{\frac{1}{\beta}} = x(\beta) < \left( \frac{4\gamma(1-x(\beta))^{\beta+\epsilon}}{4+\gamma^2 - (2-\gamma)^2(1-x(\beta))^{\beta+\epsilon}} \right)^{\frac{1}{\beta+\epsilon}}$$

$$\iff \frac{(2-\gamma)^2}{4+\gamma^2}(1-x)^\beta + \frac{4\gamma}{4+\gamma^2}\left( \frac{4+\gamma^2 - (2-\gamma)^2(1-x)^{\beta+\epsilon}}{4\gamma} \right)^{\frac{\beta}{\beta+\epsilon}} \leq 1.$$

The last inequality follows from the concavity of the function $x^{\frac{\beta}{\beta+\epsilon}}$.

Now, we will prove that the non-unit solution is decreasing in $\beta$ given that non-unit criterion holds. Notice that the non-unit solution $\tilde{\lambda}_1(\beta)$ satisfies

$$\frac{1}{n_2^\beta} - \frac{1}{(n_2 + \tilde{\lambda}_1(\beta)n_1)^\beta} = \frac{4+\gamma^2}{4\gamma}\left( \frac{1}{n_1^\beta} - \frac{1}{(n_1+n_2)^\beta} \right).$$

So, to show that $\tilde{\lambda}_1(\beta) \geq \tilde{\lambda}_1(\beta + \epsilon)$, it is sufficient to show that

$$(x^{-\beta} - \delta(1-x)^{-\beta} + \delta)^{\frac{1}{\beta}} \leq (x^{-\beta-\epsilon} - \delta(1-x)^{-\beta-\epsilon} + \delta)^{\frac{1}{\beta+\epsilon}},$$

where $x = \frac{n_2}{n_1+n_2}$ and $\delta = \frac{4+\gamma^2}{4\gamma}$. This inequality is equivalent to

$$((1-x)^\beta x^{-\beta} - \delta + \delta(1-x)^\beta)^{\frac{\beta+\epsilon}{\beta}} \leq (1-x)^{\beta+\epsilon}x^{-\beta-\epsilon} - \delta + \delta(1-x)^{\beta+\epsilon}.$$

Denote $u_0 = \left( \frac{1-x}{x} \right)^\beta$, $v_0 = (1-x)^\beta$, and $\alpha = \frac{\beta+\epsilon}{\beta}$. We get the following inequality

$$(u_0 - \delta + \delta v_0)^\alpha \leq u_0^\alpha - \delta + \delta v_0^\alpha.$$

Denote

$$f(u, v) = u^\alpha - \delta + \delta v^\alpha - (u - \delta + \delta v)^\alpha.$$

Notice that $f(u, 1) = 0$. Additionally notice that

$$\frac{\partial f}{\partial v} = \delta\alpha v^{\alpha-1} - \delta\alpha(u - \delta + \delta v)^{\alpha-1} \leq 0 \iff v \leq u - \delta + \delta v.$$

If we prove that this derivative is negative for all points $(u_0, x)$, where $x \in (v_0, 1)$, we will prove the desired result. Notice that the last inequality is the most stringent for the point $(u_0, v_0)$. At this point, we have

$$v_0 \leq u_0 - \delta + \delta v_0 \iff 0 \leq \frac{4\gamma}{n_2^\beta} - \frac{4+\gamma^2}{n_1^\beta} + \frac{(2-\gamma)^2}{(n_1+n_2)^\beta}.$$

The last inequality holds because we consider the non-unit solution.

The last property of the theorem follows from the fact that, for small ratio $\frac{n_2}{n_1}$, the non-unit criterion holds and the Taylor formula for this ratio results in the expression presented in the statement.

# B  Examples of data-sharing problem

## B.1  Taxi rides

In this section, how our framework works using motivating example from Section 3. In this example, two taxi firms use data to optimize the driver scheduling algorithm. This optimization allow them to use less driver time; thus, making their expected marginal costs lower.

We assume that the only uncertainty arising in scheduling algorithm is the duration of one kilometer ride $X \sim \mathrm{N}(\mu, 1)$. The firms do not know the parameter $\mu$. However, each firm $F_i$ has $n_i$ observations of this random variable $S_i = \{X_i^j\}_{j=1}^{n_i}$.

Their scheduling algorithm require the estimate $s$ of ride duration $X$. If the estimate is too low $s < X$, the company will need to attract more drivers and pay them more. If the estimate is too big $s > X$, the drivers will be underutilized. We assume that costs associated with these losses have MSE form $C(s, X) = a - b + b(s - X)^2$. This functional form will imply the following costs on average

$$c(s) = \mathbf{E}_X(C(s, X)) = a - b + b\, \mathbf{E}((s - X)^2) = a + b(s - \mu)^2.$$

If the firms do not collaborate, $F_i$ uses an average of its own points

$$s_{i,\mathrm{ind}} = \frac{1}{n_i} \sum_{j=1}^{n_i} X_j^i \implies s_{i,\mathrm{ind}} \sim \mathrm{N}\left(\mu, \frac{1}{n_i}\right)$$

to estimate $\mu$. This will give the following expected marginal costs

$$c_{i,\mathrm{ind}}^e = \mathbf{E}_{S_1, S_2}\, c(s_{i,\mathrm{ind}}) = a + b\, \mathbf{E}(s_{i,\mathrm{ind}} - \mu)^2 = a + \frac{b}{n_i}.$$

If the firms collaborate, they uses both samples to calculate average

$$s_{\mathrm{shared}} = \frac{1}{n_1 + n_2} \sum_{i,j} X_j^i \implies s_{\mathrm{shared}} \sim \mathrm{N}\left(\mu, \frac{1}{n_1 + n_2}\right)$$

and the expected marginal costs will be

$$c_{\mathrm{shared}}^e = \mathbf{E}_{S_1, S_2}\, c(s_{\mathrm{shared}}) = a + \frac{b}{n_1 + n_2}.$$

To solve the data-sharing problem, we need to describe the competition between firms. For simplicity we will only consider the case of Cournot competition.

As in Section 4.1.2, the firms are interested in increasing their expected profits

$$\Pi_i^e = \mathbf{E}_{S_1, S_2}(p_i q_i - c_i q_i).$$

Thus, the firms use the same quantities as in Section 4.1.2. In the case of non-collaboration, the expected profits are

$$\Pi_{i,\mathrm{ind}}^e = \left(\frac{2 - \gamma + \gamma c_{-i}^e - 2c_i^e}{4 - \gamma^2}\right)^2.$$

Similarly, in the case of collaboration,

$$\Pi_{\mathrm{shared}}^e = \left(\frac{(2 - \gamma)(1 - c_{\mathrm{shared}}^e)}{4 - \gamma^2}\right)^2.$$

The collaboration criterion is the same as in Section 4.1.2.

## B.2  Oil drilling

Assume that an oil company manager needs to optimize the costs of oil rig construction. Construction of an oil rig happens in two steps. First, geologists look at a new place for a rig and produce a noisy signal of oil presence $X$. Second, the company might try to build a rig. However, if there is no oil, the company will lose money. Thus, sometimes it is more profitable to repeat the geological expedition.

The noisy signal of geologists predicts the presence of oil in the following manner

$$\Pr(\text{success} \mid X) = \Pr(X + \epsilon \geq 0 \mid X), \ \epsilon \sim \mathrm{N}(0,1).$$

Also, the manager knows that

$$X \sim \mathrm{N}(0, \sigma^2).$$

However, the manager does not know $\sigma^2$ and needs to estimate it from the previous observations. The new expedition costs $a$, and drilling a new rig costs $b$.

The natural strategy is to repeat expeditions until the geologist find $X \geq r$ and then try to build a rig. Let $p$ be the probability of finding a good spot and $q$ be the probability of success in a good spot

$$p := \Pr(X \geq r) = \Phi\left(\frac{-r}{\sigma}\right),$$

$$q := \Pr(X + \epsilon \geq 0 \mid X \geq r) = \frac{1}{p} \int_r^\infty \int_{-x}^\infty \frac{\exp\left(\frac{-x^2}{2\sigma^2}\right)}{\sqrt{2\pi\sigma^2}} \frac{\exp\left(\frac{-y^2}{2}\right)}{\sqrt{2\pi}} \mathrm{d}y \mathrm{d}x = \frac{M(r, \sigma^2)}{p},$$

where we denote the last integral by $M$. So, the expected cost is

$$f(r) = \frac{\frac{a}{p} + b}{q} = \frac{a + \Phi\left(\frac{-r}{\sigma}\right)b}{M(r, \sigma^2)}.$$

Assume that the company estimates $\widehat{\sigma}^2$ of $\sigma^2$ and chooses $r$ according to that estimate. The loss will approximately be

$$f(r) - f(r^*) \approx f'(r^*)(r - r^*) + \frac{f''(r^*)}{2}(r - r^*)^2 = \frac{f''(r^*)}{2}(r - r^*)^2$$

$$\approx \frac{f''(r^*)}{2} r'(\sigma^2)^2 (\widehat{\sigma}^2 - \sigma^2)^2.$$

Now, if the company uses MLE to estimate $\sigma^2$, then the costs will asymptotically look like

$$f(r) \approx A + \frac{B}{n}\chi^2(1),$$

where $n$ is the number of observations.

After that the process is the same as in previous subsection.

# C   Extensions of the results

In this section, we investigate how different changes in our setting affect the results. For simplicity, we only study the case of perfect substitutes ($\gamma = 1$) and Cournot competition if not told otherwise.

## C.1   Different cost functions

In this subsection, we assume that the cost functions of the firms have the form

$$C_i(q) = c_i^e q + \frac{k}{2} q^2.$$

The profit maximization problem of the firm now has the form

$$\max_{q_i} (1 - q_i - q_{-i}) q_i - c_i^e q_i - \frac{k}{2} q_i^2.$$

Therefore, the best responses are

$$q_i^*(q_{-i}) = \frac{1 - q_{-i} - c_i^e}{2 + k}$$

and equilibrium quantities are

$$q_i^* = \frac{1 + k + c_{-i}^e - (2 + k)c_i^e}{(1 + k)(3 + k)}.$$

These equations imply the following expected profits

$$\Pi_i^e = \frac{2+k}{2}(q_i^*)^2$$

and result in the following collaboration criterion

$$\forall i \ (2+k)(n_i^{-1} - (n_1+n_2)^{-1}) > n_{-i}^{-1} - n_i^{-1}.$$

As we can see, the added non-linearity in the cost does not qualitatively change the conclusions of Theorem 5.2.

## C.2   Multiplicative substitution

In the Section 3, we assume that goods produced by the firms can be additively substituted by the outside goods (i.e., the utility function function is quasilinear). However, in general, the firms might operate in different markets with different substitution patterns.

To investigate, this question we assume the following form of utility function

$$u(q_0, q_1, q_2) = \left(q_1 + q_2 - \frac{q_1^2 + 2q_1q_2 + q_2^2}{2}\right)^\alpha q_0^{1-\alpha}.$$

($p_1 = p_2 = p$ because the goods $G_1$ and $G_2$ are perfect substitutes.) This utility prescribes the consumer to spend the share $\alpha$ of income on goods $G_1$ and $G_2$ ($B$ is small enough, so that $q_1$ and $q_2$ are much smaller than 1). So, we have the following consumer problem

$$\max_{q_1,q_2} q_1 + q_2 - \frac{q_1^2 + 2q_1q_2 + q_2^2}{2} \ \text{s.t.} \ p(q_1+q_2) \le \alpha B.$$

So, the demand is

$$p = \frac{\alpha B}{q_1 + q_2}.$$

Expected profits maximization problems are

$$\max_{q_i} \frac{\alpha B q_i}{q_1 + q_2} - c_i^e q_i.$$

The first order conditions give

$$\frac{\alpha B q_{-i}^*}{(q_1^* + q_2^*)^2} = c_i^e.$$

So, equilibrium quantities are

$$q_i^* = \frac{\alpha B q_1^* q_2^*}{c_i(q_1^* + q_2^*)^2} = \frac{\alpha B c_2^e c_1^e}{c_i^e(c_2^e + c_1^e)^2}.$$

And equilibrium profits are

$$\Pi_i^e = \frac{\alpha B (c_{-i}^e)^2}{(c_1^e + c_2^e)^2} = \frac{\alpha B}{\left(1 + \frac{c_i^e}{c_{-i}^e}\right)^2}.$$

This formula imply the following collaboration criterion

$$\forall i \ 1 + \frac{c_{i,\text{ind}}^e}{c_{-i,\text{ind}}^e} < 1 + \frac{c_{\text{share}}^e}{c_{\text{share}}^e} = 2.$$

As can be seen, the collaboration is always not profitable for the firm with more data.

### C.3 Asymmetric costs

In this subsection, we assume that firms have asymmetric cost functions

$$c_i^e(n) = a_i + \frac{b_i}{n^{\beta_i}},$$

where $c_i^e(n)$ is the expected marginal cost of the company if it use $n$ points for training. Section 4.1.2 gives the following the expected profits

$$\Pi_i^e = \left( \frac{2 - \gamma - 2c_i^e + \gamma c_{-i}^e}{4 - \gamma^2} \right)^2.$$

As in Section 4.1.2, the companies will collaborate only if both of them profit from collaboration

$$\forall i \; \Pi_{\text{shared}}^e > \Pi_{i,\text{ind}}^e \iff 2b_i(n_i^{-\beta_i} - n_{\text{shared}}^{-\beta_i}) > \gamma b_{-i}(n_{-i}^{-\beta_{-i}} - n_{\text{shared}}^{-\beta_{-i}}).$$

Denote $f_i(n_1, n_2) = 2b_i(n_i^{-\beta_i} - n_{\text{shared}}^{-\beta_i}) - \gamma b_{-i}(n_{-i}^{-\beta_{-i}} - n_{\text{shared}}^{-\beta_{-i}})$. This function can be interpreted as firm $F_i$ reediness to collaborate: the higher this value is the more firm $F_i$ wants to collaborate with its competitor $F_{-i}$.

**Theorem C.1.** *Assume the setting described above. The function $f_i(n_1, n_2)$ has the following properties:*

1. *$f_i(n_1, n_2)$ is decreasing in $\gamma$*

2. *$f_i(n_1, n_2)$ is decreasing in $\beta_i$ and increasing in $\beta_{-i}$ if $\forall i \; \beta_i \ln n_i > 1$.*

3. *$f_i(n_1, n_2)$ is increasing in $b_i$ and decreasing in $b_{-i}$.*

*Proof.* The first property is evident

$$\frac{\partial f_i}{\partial \gamma} = -b_{-i}(n_i^{-\beta_{-i}} - n_{\text{shared}}^{-\beta_{-i}}) < 0.$$

We can prove the last property similarly

$$\frac{\partial f_i}{\partial b_i} = 2(n_i^{-\beta_i} - n_{\text{shared}}^{-\beta_i}) > 0,$$

$$\frac{\partial f_i}{\partial b_{-i}} = -\gamma(n_i^{-\beta_{-i}} - n_{\text{shared}}^{-\beta_{-i}}) < 0.$$

The first part of the second property follows from the following

$$\frac{\partial f_i}{\partial \beta_i} = 2b_i \left( \frac{\ln n_i}{n_{\text{shared}}^{\beta_i}} - \frac{\ln n_i}{n_i^{\beta_i}} \right).$$

Notice that the function $g(x) = \frac{\ln x}{x^{\beta_i}}$ is decreasing in $x$ on $x > e^{1/\beta_i}$ because

$$\frac{\partial g}{\partial x} = \frac{1 - \beta_i \ln x}{x^{\beta_i + 1}} < 0.$$

Therefore, $\frac{\partial f_i}{\partial \beta_i} < 0$: $f_i$ is decreasing in $\beta_i$. The second part can be proved similarly. $\square$

The first property is similar to the first property in Theorem 5.2. The firms with more similar goods have less incentives to collaborate.

The last two properties show that the firm wants to collaborate with its competitor when the firm's machine learning model is bad ($\beta_i$ is low or $b_i$ is high) or the competitor's model is good ($\beta_{-i}$ is high or $b_{-i}$ is low).

## C.4 Heterogeneity

In this section, we solve the same model as in Appendix B, but allowing for heterogeneity between firms. Each firm will need to optimize the MSE costs

$$c_i(s) = a - b + b\,\mathbf{E}((s - X_i)^2) = a + b(s - \mu_i)^2, \; X_i \sim \mathrm{N}(\mu_i, 1).$$

On the contrary to the Appendix B, the means of the noise will be different among firms and will be drawn at the start of the game from the distribution

$$\mu_i \sim \mathrm{N}(\mu, \sigma^2).$$

We assume that $\mu$ is unknown and $\sigma^2$ is known.

When firms do not collaborate, the expected costs are the same as in Appendix B. When firms collaborate, MLE for the means are

$$s_{i,\text{shared}} = \frac{2\sigma^2 n_1 n_2 Y_i + n_i Y_i + n_{-i} Y_{-i}}{2\sigma^2 n_1 n_2 + n_1 + n_2},$$

where $n_i$ is the number of data points of firm $F_i$, $Y_i$ is the average of noise. Then expected marginal costs will be

$$c_{i,\text{shared}}^e = a + b\,\mathbf{E}_{S_1,S_2}((s_{i,\text{shared}} - \mu_i)^2).$$

By substitution,

$$\mathbf{E}((s_{i,\text{shared}} - \mu_i)^2) =$$
$$\mathbf{E}\left(\frac{n_{-i}^2(\mu_1 - \mu_2)^2}{(2\sigma^2 n_1 n_2 + n_1 + n_2)^2} + \frac{(2\sigma^2 n_{-i} + 1)^2 n_i}{(2\sigma^2 n_1 n_2 + n_1 + n_2)^2} + \frac{n_{-i}}{(2\sigma^2 n_1 n_2 + n_1 + n_2)^2}\right)$$
$$= \frac{2\sigma^2 n_{-i} + 1}{2\sigma^2 n_1 n_2 + n_1 + n_2}.$$

Therefore, $c_{i,\text{shared}}^e$ is increasing in $\sigma^2$: the more different the firms are, the less collaboration gives. Two corner cases give

$$\sigma^2 = 0 \implies c_{i,\text{shared}}^e = a + \frac{b}{n_1 + n_2},$$
$$\sigma^2 \to \infty \implies c_{i,\text{shared}}^e \to a + \frac{b}{n_i}.$$

However, notice that collaboration criteria in the Cournot case does not depend on $\sigma^2$ and hence does not change

$$\Pi_{i,\text{share}}^e - \Pi_{i,\text{ind}}^e \geq 0 \iff 2 - \gamma - 2c_{i,\text{share}}^e + \gamma c_{-i,\text{share}}^e \geq 2 - \gamma - 2c_{i,\text{ind}}^e + \gamma c_{-i,\text{ind}}^e$$
$$\iff -\frac{2(2\sigma^2 n_{-i} + 1)}{2\sigma^2 n_1 n_2 + n_1 + n_2} + \frac{\gamma(2\sigma^2 n_i + 1)}{2\sigma^2 n_1 n_2 + n_1 + n_2} \geq -\frac{2}{n_i} + \frac{\gamma}{n_{-i}} \iff 0 \geq -2\frac{n_{-i}}{n_i} + \gamma\frac{n_i}{n_{-i}}.$$

# D  Other models of coalition formation

In this section, we present models of coalition formation different form the model presented in Section 7.

## D.1  Cooperative data sharing

We use an solution concept motivated by the $\alpha$-core (Von Neumann & Morgenstern, 1944), which is standard in cooperative game theory. The $\alpha$-core consists of partitions that are incentive-compatible, ensuring that no subset of firms wants to deviate. Incentive compatibility indicates that these partitions more stable and likely to occur.

First, we introduce a modification of $\alpha$-core, tailored to our setup. This deviation is necessary due to two reasons, non-transferable utility and the presence of externalities (dependencies of utilities on the

actions of all participants), which arise in our setting because all firms compete with each other on the market.

Let $N = \{F_1, \ldots, F_m\}$ be the set of all firms. A partition of $N$ is a set of subsets $\{S_1, \ldots, S_k\}$ : $S_i \subseteq N$, such that $\forall i, j, i \neq j : S_i \cap S_j = \emptyset$ and $\cup_i S_i = N$. We denote the set of all partitions of $N$ as $P(N)$.

**Definition D.1** ($\alpha$-Core). A partition $P$ belongs to the $\alpha$-core of the game if and only if

$$\nexists S \subseteq N : \forall F \in S, Q \in P(N \setminus S) \quad \Pi_F^e(P) < \Pi_F^e(Q \cup \{S\}),$$

where $\Pi_F^e(P)$ is expected profit of the firm $F$ if market coalition structure is $P$.

Intuitively, a partition $P$ belongs to the $\alpha$-core if no set of companies wants to deviate from this partition. Here we say that a subset of companies wants to deviate, if any company in this subset will increase its profits by joining this new coalition, regardless of how the remaining companies split into groups.

In the next theorem we identify one coalition structure that always belongs to the $\alpha$-core of the data-sharing game, demonstrating that the not-emptiness of the core.

**Theorem D.2.** *W.l.o.g. assume that $n_1 \geq n_2 \geq \cdots \geq n_m$, where $n_i$ is the number of data points of firm $F_i$. Consider partitions $P_i = \{A_i, N \setminus A_i\}$, where $A_i = \{F_1, F_2, \ldots, F_i\}$. Let $i^* = \arg\max_i \Pi_{F_1}^e(P_i)$. Then $P_{i^*}$ belongs to $\alpha$-core.*

Before we start to prove this theorem, we formulate the following lemma.

**Lemma D.3.** *Let $Q \subset N$ and $F \in Q$. Then*

$$\Pi_F^e(\{Q, N \setminus Q\}) = \frac{(2 - \gamma - (2 + \gamma(m - 1 - |Q|)) c_Q^e + \gamma(m - |Q|) c_{N \setminus Q}^e)^2}{(2 - \gamma)^2 (2 + (m - 1)\gamma)^2},$$

*where $c_X^e = a + b/n_X^\beta$, expected marginal cost of coalition $X$, and $n_X = \sum_{F \in X} n_F$, the number of data points of coalition $X$.*

*Proof.* The results of Section 4.1.2 give the following expected profits

$$\Pi_i^e = \frac{(2 - \gamma - (2 + \gamma(m - 2)) c_i^e + \gamma \sum_{j \neq i} c_j^e)^2}{(2 - \gamma)^2 (2 + (m - 1)\gamma)^2}.$$

In our case,

$$c_i = \begin{cases} c_Q, & F_i \in Q, \\ c_{N \setminus Q}, & F_i \notin Q, \end{cases}$$

By substituting $c_i^e$ into the profit equation for $F \in Q$, we get

$$\Pi_F^e(\{Q, N \setminus Q\}) = \frac{(2 - \gamma - (2 + \gamma(m - 2)) c_Q^e + \gamma(|Q| - 1) c_Q^e + \gamma(m - |Q|) c_{N \setminus Q}^e)^2}{(2 - \gamma)^2 (2 + (m - 1)\gamma)^2}$$

$$= \frac{(2 - \gamma - (2 + \gamma(m - 1 - |Q|)) c_Q^e + \gamma(m - |Q|) c_{N \setminus Q}^e)^2}{(2 - \gamma)^2 (2 + (m - 1)\gamma)^2}.$$

$\square$

**Corollary D.4.** *Let $Q, Q' \subset N$, $F \in Q$, and $F' \in Q$. Then*

$$\Pi_F^e(\{Q, N \setminus Q\}) \geq \Pi_{F'}^e(\{Q', N \setminus Q'\}) \iff$$

$$\frac{2 + \gamma(m - 1 - |Q'|)}{n_{Q'}^\beta} - \frac{\gamma(m - |Q'|)}{n_{N \setminus Q'}^\beta} \geq \frac{2 + \gamma(m - 1 - |Q|)}{n_Q^\beta} - \frac{\gamma(m - |Q|)}{n_{N \setminus Q}^\beta}.$$

Now, we are ready to prove Theorem D.2.

*Proof.* We prove the theorem by contradiction. Suppose that there exists a coalition $Q$ for which deviation is profitable.

First, we consider the case $A_{i^*} \cap Q \neq \emptyset$. Let $F \in A_{i^*} \cap Q$. Notice that

$$\Pi_F^e(P_{i^*}) = \Pi_{F_1}^e(P_{i^*}) \geq \Pi_{F_1}^e(P_{|Q|}) \geq \Pi_F^e(\{Q, N \setminus Q\}).$$

The first equality follows from Lemma D.3. The first inequality follow from the definition of $i^*$. The last inequality follows from Corollary D.4 and observation $n_{A_{|Q|}} \geq n_Q$. So, we get a contradiction. Thus, $A_{i^*} \cap Q = \emptyset$.

Let $F \in Q$. In this case,

$$\Pi_F^e(P_{i^*}) \geq \Pi_F^e(\{Q, N \setminus Q\}).$$

This inequality follows from Corollary D.4, the fact that $n_{N \setminus A_{i^*}} \geq n_Q$, and observation $|N \setminus A_{i^*}| \geq |Q|$. This contradiction proves the theorem. $\square$

In the partition described in the theorem, the firms form two coalitions: one contains the $i^*$ companies with the largest datasets and the other one contains all other firms and leads to largest profits for the company with the largest amount of data. However, it may not be the only one in the core since the $\alpha$-core is one of the most permissive cores in the cooperative game literature: a set of companies will deviate only if it is better-off irrespective of the actions of others, sometimes resulting non-economically plausible partitions.

The following example illustrates this flaw. Assume that there are three firms in the market and $n_1 \gg n_2 \gg n_3$. In this case, the $\alpha$-core consists of two equilibria: $\{\{F_1, F_2\}, \{F_3\}\}$ and $\{\{F_1\}, \{F_2\}, \{F_3\}\}$, but the first equilibrium is less economically plausible. Indeed, for the first firm, the second equilibrium is more profitable than the first. However, this firm will not deviate because of being afraid that the second firm might collaborate with the third. Nevertheless, this fear is ungrounded since the second firm expects less profit from the collaboration with the third, than from staying alone.

## D.2 Universal data-sharing treaty

The first model consider a following situation. Imagine that companies might sign some treaty that will their data accessible for use for other companies. To make companies comply to the outcome, the decision is made by consensus. In terms of coalition formation game, each company either agree or disagree to participate in grand coalition. If any company disagrees, the grand coalition is not formed and companies act like singletons.

W.l.o.g., assume that $n_1 \geq n_2 \geq \cdots \geq n_m$. The grand coalition will be Nash equilibrium in this game only if all firms will be better off in the grand coalition

$$\forall F \ \Pi_F^e(\{\{F_1, F_2, \ldots, F_m\}\}) > \Pi_F^e(\{\{F_1\}, \{F_2\}, \ldots, \{F_m\}\}).$$

Since grand coalition profit is the same for all firms and singleton profit is biggest for the firm with the most amount of data, this system of inequalities is equivalent to the inequality for the first firm

$$\Pi_{F_1}^e(\{\{F_1, F_2, \ldots, F_m\}\}) > \Pi_{F_1}^e(\{\{F_1\}, \{F_2\}, \ldots, \{F_m\}\}).$$

By substitution, this inequality takes the form

$$\frac{\gamma m - m - 1}{n^\beta} > -\frac{m+1}{n_1^\beta} + \sum_{j=1}^m \frac{\gamma}{n_j^\beta},$$

where $n = \sum_j n_j$.

## D.3 Data-sharing treaty

The second model is similar to the first, but here the members of the treaty share data only between themselves. Thus, not everybody needs to agree to build a coalition and resulting structure will be one coalition with several members and singleton coalitions. Formally, we assume the following game. Each firm has two actions: Y and N. All firms that answer Y form a coalition between themselves and all firms that answer N form singleton coalitions. We are interested in Nash equilibria of this game.

W.l.o.g., assume that $n_1 \geq n_2 \geq \cdots \geq n_m$. The following lemma is useful for the description of all equilibria of this game

**Lemma D.5.** *Let $S$ be a set of firm that answer Y in equilibrium and $i^* = \min\{i \mid F_i \in S\}$. Then $\forall j \geq i^* \; F_j \in S$.*

*Proof.* To show this it is sufficient to show that a single firm $F$ always want to join a coalition $S$ that has more data than the firm. This is equivalent to the following inequality

$$\Pi_F^e(\{S \cup \{F\}\} \cup \{\{G\} \mid G \notin S \cup \{F\}\}) > \Pi_F^e(\{S\} \cup \{\{G\} \mid G \notin S\})$$

$$\iff \frac{\gamma(|S|+1) - m - 1}{(n_S + n_F)^\beta} > \frac{\gamma|S|}{n_S^\beta} - \frac{m+1-\gamma}{n_F^\beta},$$

where $n_S = \sum_{G \in S} n_G$. Denote $y := \frac{n_F}{n_S}$. The last inequality is equivalent to

$$(m+1-\gamma)\left(1 - \frac{y^\beta}{(1+y)^\beta}\right) > \gamma|S|\left(y^\beta - \frac{y^\beta}{(1+y)^\beta}\right).$$

Since $\gamma < 1$ and $|S| < m$, we have $m + 1 - \gamma > \gamma|S|$. Furthermore, $1 - \frac{y^\beta}{(1+y)^\beta} \geq y^\beta - \frac{y^\beta}{(1+y)^\beta} > 0$ because $1 \geq y > 0$. Therefore, the inequality above holds. $\qquad\square$

Lemma D.5 greatly constraints possible equilibria of the game. Indeed, the equilibrium set of companies that say Y may only have the form $S_i = \{F_j \mid j \geq i\}$. To check whether $S_i$ appears in the equilibrium, we need to check the following system of inequalities

$$\forall j < i \; \Pi_{F_j}^e(\{S_i\} \cup \{\{G\} \mid G \notin S_i\}) \geq \Pi_{F_j}^e(\{S_i \cup \{F_j\}\} \cup \{\{G\} \mid G \notin S_i \cup \{F_j\}\}),$$

$$\forall j \geq i \; \Pi_{F_j}^e(\{S_i\} \cup \{\{G\} \mid G \notin S_i\}) \leq \Pi_{F_j}^e(\{S_i \cup \{F_j\}\} \cup \{\{G\} \mid G \notin S_i \cup \{F_j\}\}).$$

These inequalities are closest for the firms around threshold $i$. Thus, this system is equivalent to two inequalities

$$\Pi_{F_{i-1}}^e(\{S_i\} \cup \{\{G\} \mid G \notin S_i\}) \geq \Pi_{F_{i-i}}^e(\{S_i \cup \{F_{i-1}\}\} \cup \{\{G\} \mid G \notin S_i \cup \{F_{i-1}\}\}),$$

$$\Pi_{F_i}^e(\{S_i\} \cup \{\{G\} \mid G \notin S_i\}) \leq \Pi_{F_i}^e(\{S_i \cup \{F_i\}\} \cup \{\{G\} \mid G \notin S_i \cup \{F_i\}\}).$$

# E   Welfare analysis

In this section, we consider how collaboration affects the cumulative utility of firms and consumers. We start with the definition of welfare.

**Definition E.1.** The sum of consumer surplus and firms surplus $W$ is called welfare. In our term this quantity is equal to the sum of consumers utility and firms profits

$$W := u + \sum_{i=1}^{m} \Pi_i.$$

The definition of welfare in our case is simpler than in general case because consumers have quasilinear preferences.

Welfare is an important quantity in policy analysis. Welfare reflects the cumulative benefits of all market participants. Thus, if redistribution between people is possible (e.g., through government transfers), increase in welfare means increase in utility for all people given the right redistribution scheme.

Now, we want to evaluate how data-sharing in duopoly case affects the expected welfare. This will help us to understand in which cases the firms should be incentivized to collaborate with each other. By direct substitution, we get

$$W^e = q_1^* + q_2^* - \frac{(q_1^*)^2 + (q_2^*)^2 + 2\gamma q_1^* q_2^*}{2} + B - c_1^e q_1^* - c_2^e q_2^*.$$

We show that collaboration increases welfare in the Cournot case if the marginal costs of the firms are close enough: $|c_1^e - c_2^e| \leq \frac{\gamma}{6}$. To accomplish that, we will first demonstrate that $\frac{\partial W}{\partial c_i^e} < 0$ if $c_i^e > c_{-i}^e$ and then show that $W$ decreases along the line $c_1^e = c_2^e = c$. These properties will show

that collaboration increases welfare since after collaboration the costs of the companies equalizes and become smaller.

The proof of the first part follows from the derivations below. W.l.o.g. assume that $n_1 \geq n_2$, then $c_1^e \leq c_2^e$. Direct computations show

$$\frac{\partial W^e}{\partial c_2^e} = \frac{\partial q_1^*}{\partial c_2^e}(1 - c_1^e) + \frac{\partial q_2^*}{\partial c_2^e}(1 - c_2^e) - \frac{\partial q_1^*}{\partial c_2^e}(q_1^* + \gamma q_2^*) - \frac{\partial q_2^*}{\partial c_2^e}(q_2^* + \gamma q_1^*) - q_2^*.$$

The results of Section 4.1.2 give

$$q_i^* = \frac{2 - \gamma - 2c_i^e + \gamma c_{-i}^e}{4 - \gamma^2}.$$

Substituting the expressions above, we have

$$\frac{\partial W^e}{\partial c_2^e} = \frac{(12 - \gamma^2)c_2^e - (8\gamma - \gamma^3)c_1^e - (12 - 8\gamma - \gamma^2 + \gamma^3)}{(4 - \gamma^2)^2}$$

$$< \frac{(12 - \gamma^2)c_2^e - (8\gamma - \gamma^3)\left(c_2^e - \frac{\gamma}{6}\right) - (12 - 8\gamma - \gamma^2 + \gamma^3)}{(4 - \gamma^2)^2}$$

$$= \frac{-(12 - 8\gamma - \gamma^2 + \gamma^3)(1 - c_2^e) + \frac{\gamma}{6}(8\gamma - \gamma^3)}{(4 - \gamma^2)^2}.$$

The restriction $c_2 \leq 1 - \frac{\gamma}{2}$ gives

$$\frac{\partial W^e}{\partial c_2^e} \leq \frac{-(12 - 8\gamma - \gamma^2 + \gamma^3)\frac{\gamma}{2} + \frac{\gamma}{6}(8\gamma - \gamma^3)}{(4 - \gamma^2)^2}$$

This inequality is equivalent to the following inequality

$$36 - 24\gamma - 3\gamma^2 + 3\gamma^3 - 8\gamma + \gamma^3 > 0 \iff (36 - 3\gamma^2)(1 - \gamma) + \gamma^3 > 0.$$

The last inequality holds since $\gamma < 1$.

The proof of the second part comes from the computations below. Assume $c_1^e = c_2^e = c$, we want to show that $W$ decreases in $c$. We have

$$W^e = \frac{(12 - 8\gamma - \gamma^2 + \gamma^3)c^2 - (24 - 16\gamma - 2\gamma^2 + 2\gamma^3)c + 12 - 8\gamma - \gamma^2 + \gamma^3}{(4 - \gamma^2)^2} + B$$

$$= \frac{(12 - 8\gamma - \gamma^2 + \gamma^3)(1 - c)^2}{(4 - \gamma^2)^2} + B.$$

Clearly, this function is decreasing in $c$ for $c < 1$.

# F   Additional experiments for Section 7

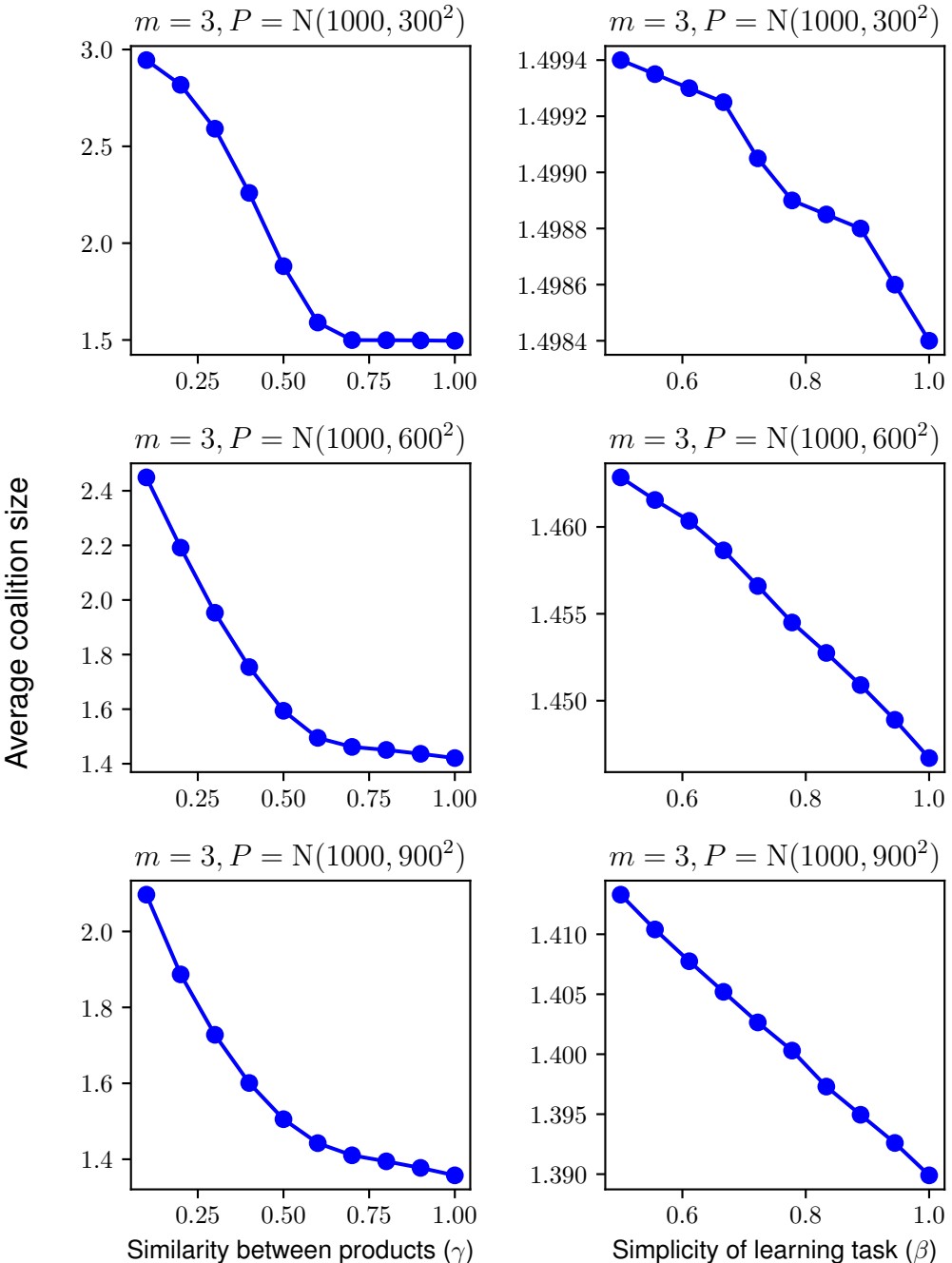

Figure 2: The dependence of the average coalition size on $\gamma$ and $\beta$ for in synthetic experiments. The $y$-axes report the mean of the average size of the coalitions in the equilibrium partition, where mean is taken over 10000 Monte Carlo simulations of the game. The firms' dataset sizes are sampled from clipped (at 1) distribution $P$. There are 3 firms in the market engaging in the Cournot competition. The default values of $\gamma$ and $\beta$ are equal to 0.8 and 0.9, respectively.

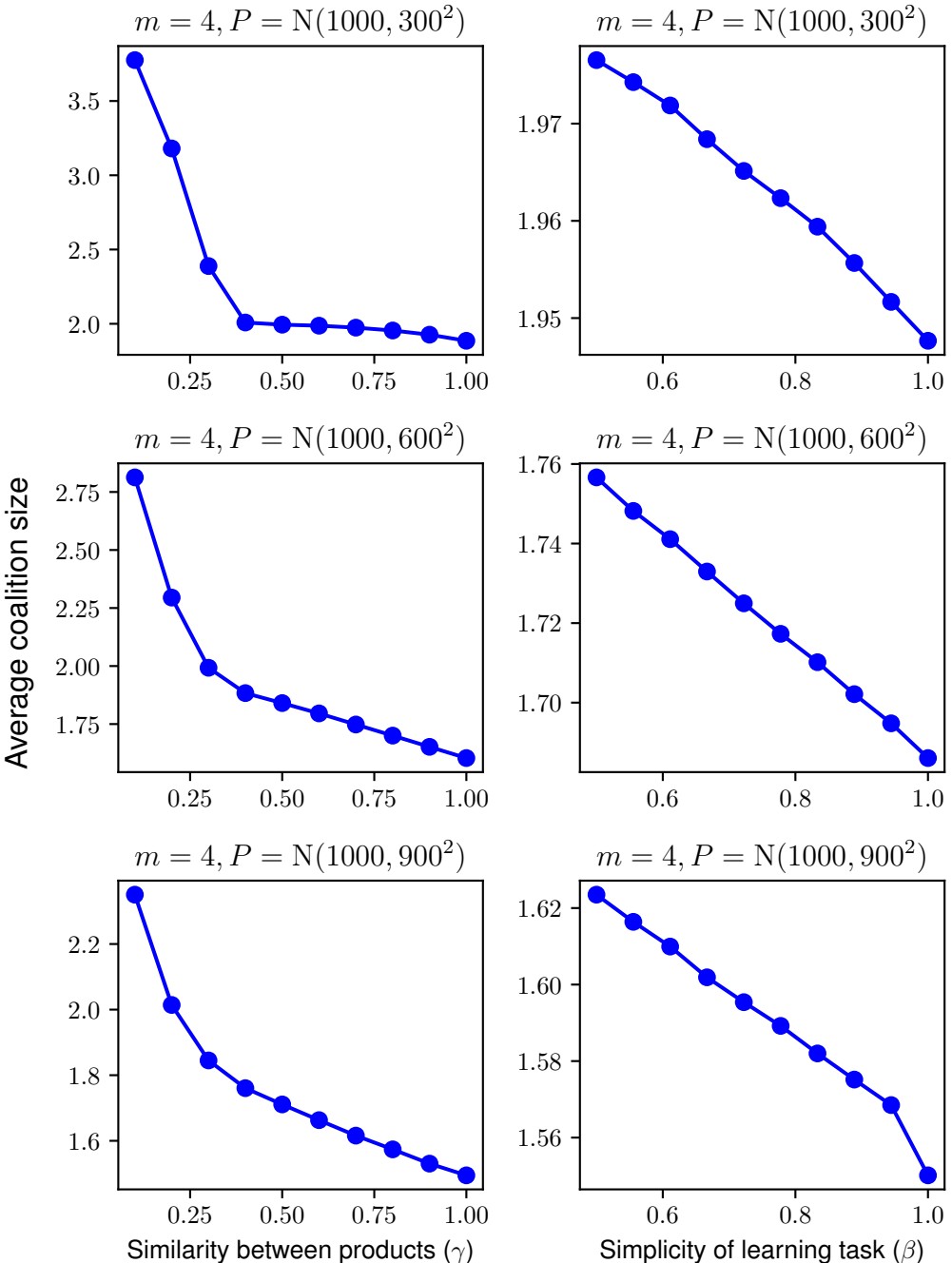

Figure 3: The dependence of the average coalition size on $\gamma$ and $\beta$ for in synthetic experiments. The $y$-axes report the mean of the average size of the coalitions in the equilibrium partition, where mean is taken over 10000 Monte Carlo simulations of the game. The firms' dataset sizes are sampled from clipped (at 1) distribution $P$. There are 4 firms in the market engaging in the Cournot competition. The default values of $\gamma$ and $\beta$ are equal to 0.8 and 0.9, respectively.

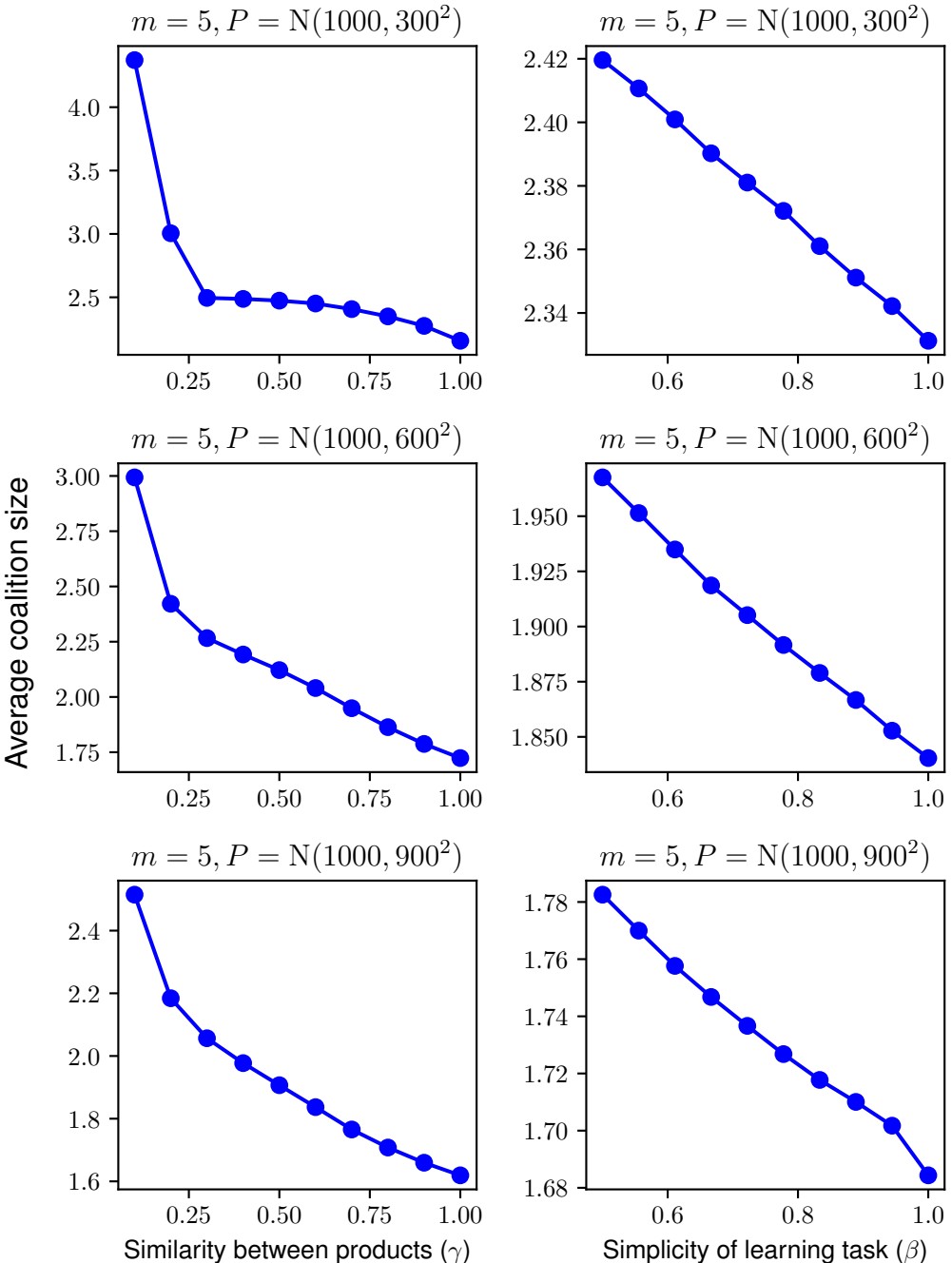

Figure 4: The dependence of the average coalition size on $\gamma$ and $\beta$ for in synthetic experiments. The $y$-axes report the mean of the average size of the coalitions in the equilibrium partition, where mean is taken over 10000 Monte Carlo simulations of the game. The firms' dataset sizes are sampled from clipped (at 1) distribution $P$. There are 5 firms in the market engaging in the Cournot competition. The default values of $\gamma$ and $\beta$ are equal to 0.8 and 0.9, respectively.

