# OpenReview forum: "Strategic Data Sharing between Competitors"
_NeurIPS.cc/2023/Conference — NeurIPS 2023 poster_

### Official Review · Reviewer_JkGv · 2023-07-03

**Soundness:** 4 excellent
**Presentation:** 3 good
**Contribution:** 3 good
**Rating:** 8
**Confidence:** 4

**Summary:**

This paper uses algorithmic game theory to study potential data sharing between competing firms that train machine learning models with similar goals. The authors propose a new framework that can be used to analyze trade-offs faced by data-using agents who make decisions about whether to collaborate (via data sharing) to improve their own ML performance, while potentially losing out on profits because they also improved their competitors ML performance.

The authors draw on conventional market models from academic economics to make predictions about how different structural factors might impact collaboration decisions. This analysis has implications for the development of regulations and norms pertaining to data sharing.


**Strengths:**

This paper has a number of strengths. The motivation, combination of theoretical arguments with simulation evidence, and reasonable use of assumptions all stood out.

First, the paper’s motivation is quite strong. The authors argue that the incentives underlying data sharing between machine learning operating firms is understudied, and that using the lens of theoretical modeling can highlight how market conditions might impact sharing behaviors.

The work combines theory (drawing on published economics literature) and a simulation experiment. I thought this combination was convincing: the experimental component is likely to help readers understand the implications of the theoretical framework.

There are a lot of assumptions at play, but they seem fairly plausible on the whole. Thinking in terms of whether this theory-focused paper could guide practical decision-making (by firms) or policy making (by regulators), the current draft devotes enough attention to arguing for plausibility. I do think there’s room to see more discussion of where these assumptions are more or less plausible (see below), though this may be out of scope for a theory paper that’s trying to propose a new framework and ask for some re-thinking.


**Weaknesses:**

In my view, the main threat to the validity and impact of the work is the reliance on assumptions from conventional economics. The authors are very upfront about signposting which prior works are most foundational (e.g. the 1979 work defining representative consumer with quasi-linear quadratic utility, and classical models of competition that focus on quantities vs. prices) but empirical validation could help this line of work have more impact in the long-term.

Put another way, the current draft does a great job of pointing out relevant literature that it builds off, but could do more to explicitly state why certain assumptions fit a specific empirical context. Of course, the authors may not wish to zoom in on a specific context, which is a fair choice for scoping the paper.

The paper also discusses data as a “key asset” (e.g. in the Introduction) in a very broad sense. None of the claims are situated relative to specific use cases of data and ML. Examples like ad tech and a “production process” are gestured at, and of course the running example of taxi driver scheduling is helpful. On the whole, however, I think it may be hard for readers to assess how contextually dependent some of the claims and results are. Overall, the impact of the work could be improved by clarifying the extent to which these analyses are or are not contextually dependent (even in terms of specific numerical characterizations of data scaling behavior / “data impact model”).


**Questions:**

Early in the Introduction, free-riding and non-collaboration concerns are mentioned. The authors may want to engage briefly with sociological work on collective action (though perhaps this better suited for future work and out of scope for this paper). I think a big open question for future work along these lines is which types of data are best handled with an “economic-leaning” model vs a “sociological-leaning” model that includes non-monetary incentives facing individual data generating agents.

In general, the framework also seems to lack any notion of the possibility of commons or public goods. I could imagine adding an additional stage (or several) to the game to account for this. Fully engaging with this topic is almost certainly out of scope given the current space constraints, but perhaps worth a brief mention.

While I expect the core audience for this kind of paper (i.e. readers familiar with some of the references already or interested in data collection games) will follow most sections, there’s potential to strengthen the broader impact of the paper just by adding a bit more high-level summarization of each sections.. Sections 3.1 and the end of Section 5 do a nice job of this kind of discussion, but there’s opportunities to emphasize this kind of recap in other sections.

As a minor comment, it may help the paper to discuss whether data-dependent products lend themselves to Bertrand vs. Cournot competition. It seems Cournot may be preferred, but I didn’t quite understand why. This relates to my main high-level comment about the work, which is that I think readers will want to know how different assumptions here map to different data-dependent technology contexts.


**Limitations:**

The work is reasonable in discussing its own limitations.

In terms of societal implications, I think the discussion in the paper is commensurate to potential concerns. Overall, the paper is a bit light on discussion of how data sharing would work in practice (not touching on topics like public goods, anti-trust concerns, consumer welfare, etc.) but I think this is reasonable given space constraints and paper scope.

As noted above, I think some engagement with literature on collective action, commons, and public goods could be insightful, or even a brief mention of why the authors think the collective action / commons perspective on data-dependent technologies is not relevant.

---

> ### Author Rebuttal · Authors · 2023-08-09
>
> Thank you for your valuable and constructive feedback. In the following we address the weaknesses and questions raised in your review.
>
> **Weaknesses**
>
> **In my view, the main threat to the validity and impact of the work is the reliance on assumptions from conventional economics … The current draft … could do more to explicitly state why certain assumptions fit a specific empirical context. Of course, the authors may not wish to zoom in on a specific context, which is a fair choice for scoping the paper.**
>
> Thank you for your comments. We certainly agree that designing and validating market and data impact models for specific applications is important for the framework's applicability, and we deem this direction interesting for future work. The current paper indeed stays away from focusing on a specific real-world application: please refer to our general response for a thorough justification on this matter.
>
> **The paper also discusses data as a "key asset" (e.g. in the Introduction) in a very broad sense. None of the claims are situated relative to specific use cases of data and ML … Overall, the impact of the work could be improved by clarifying the extent to which these analyses are or are not contextually dependent**
>
> Thank you for your valuable suggestion. Indeed, the parts of the framework may vary depending on the context, due to the generality of the framework. We plan to provide further explanation on the framework adaptation by giving more specific examples of relevant industries (also suggested by Reviewer 4Akt) and discussing example numerical characterizations of our models (also suggested by Reviewer Jw59).
>
> **Questions**
>
> **The authors may want to engage briefly with sociological work on collective action (though perhaps this better suited for future work and out of scope for this paper). I think a big open question for future work along these lines is which types of data are best handled with an "economic-leaning" model vs a "sociological-leaning" model that includes non-monetary incentives facing individual data generating agents.**
>
> Thank you for your valuable suggestion. We certainly agree that analyzing the differences between economic and sociological modeling of data sharing is an exciting direction for future work. We believe that our framework can capture other (e.g., sociological) data-sharing approaches by adapting the collaboration scheme. We see this as orthogonal to the focus of the current work, which studies non-cooperative games since they align with the classic economic modeling of firms as rational, profit-driven, and self-interested. However, we will be happy to add a discussion on possible sociological extensions as interesting future work.
>
> In fact, Appendix E provides some evidence about the relevance of collective action theory toward finding widely beneficial schemes for data sharing. This section analyzes global welfare in the context of the data-sharing problem. The result suggests that the welfare is maximized when data sharing occurs (in the case of full data sharing between two firms).
>
> **In general, the framework also seems to lack any notion of the possibility of commons or public goods. I could imagine adding an additional stage (or several) to the game to account for this. Fully engaging with this topic is almost certainly out of scope given the current space constraints, but perhaps worth a brief mention.**
>
> We certainly agree that the public goods perspective might be relevant here since data is non-rival. However, data does not seem to be the commons because it belongs to either companies or individuals in most legal systems. At the same time, machine learning models and data do not seem to be public goods since they are excludable. Therefore, we are unsure how to include the discussion of public goods in our framework. That said, one aspect that may be possible to model is the creation of public datasets (e.g., ImageNet).
>
> **There's potential to strengthen the broader impact of the paper just by adding a bit more high-level summarization of each sections.**
>
> Thank you for the valuable suggestion. We will seek to provide further summaries and intuition for the next version of the paper.
>
> **It may help the paper to discuss whether data-dependent products lend themselves to Bertrand vs. Cournot competition. It seems Cournot may be preferred, but I didn't quite understand why. I think readers will want to know how different assumptions here map to different data-dependent technology contexts.**
>
> We focus on the Bertrand and Cournot competition because these models are the most popular in empirical research and are a basis for all other competition models. In some cases, we only provided the analysis for the Cournot model since the derivations were less tedious with the parametrization of the market model adopted in the paper. However, we expect these results to also transfer to the Bertrand model.
>
> Regarding the mapping to different contexts, completely answering this question is beyond speculation on our side, requiring a separate economic study since we are not aware of any current economic work that empirically studies the question of data sharing for machine learning (see also our main response). We hope to discuss how one could adopt our framework to real-world contexts in the next version of our paper, for instance by covering examples of relevant industries and numerical characterizations of our models, as mentioned above.

---

> > ### Comment · Reviewer_JkGv · 2023-08-11
> > **Rebuttal helps address areas for improvement**
> >
> > Thanks for this rebuttal, authors! Overall, I think the points raised here are fair (e.g., challenges with incorporating public goods directly into this current work, the discussion of using global welfare lens).
> >
> > I thought the general argument for avoid hyper-specific data use-case here is fair as well.
> >
> > Overall, I appreciate this additional information from the authors, and it sounds like it will be possible to build on some of the areas for improvement in camera ready. I think this paper will be a valuable contribution to the conference.

---

> > > ### Author Response · Authors · 2023-08-15
> > > **Thank you for your response!**
> > >
> > > Thank you for your timely response! We appreciate your constructive and positive feedback, which we will incorporate in the next version of the paper. In particular, we will discuss the non-rivalry of data and the obstacles in front of collective action in collaborative learning, as well as the positioning of this work as a general framework that opens the door for application-specific models.

---

### Official Review · Reviewer_DZoM · 2023-07-03

**Soundness:** 3 good
**Presentation:** 4 excellent
**Contribution:** 2 fair
**Rating:** 6
**Confidence:** 5

**Summary:**

The authors present a trainable model for optimising the benefits of data sharing based valuation of data, benefits of the shared data sets for the inference model, in a conventional market model.  The authors point out that the model can be used to find an optimal data sharing strategy.

The model is nicely developed and mathematically sound, presented clearly in an understandable manner. It states the assumptions that are used in the development of the model and provides a good a good summary of the consequences, the effect of the simplicity of the learning task, and similarity of the products build using the shared data sets. The mean coalition size is shown as function of these parameters.

**Strengths:**

The paper is written and goes through the relevant steps in creation of the model.  It provides excellent argumentation and conclusions from the assumptions made.

**Weaknesses:**

The biggest weakness I see in the paper is its assumption that the collaborative partners share the same data distribution and the data is modelled as iid samples of a common distribution.  This is generally the situation when there less sense of sharing data, as one has the capability, in time,  to  the get a representative data set that is enough for an highly accurate inferred model.

However, the real problem, is in a case that the data sharing partners work with differently distributed,  where the parties are do not have the capability to reproduce the data that the other parties have. This is for example the case where medical X-rays are taken with different X-ray machines, in diverse set of hospitals and the aim is to build an inference model that works generally with a variable types of machines.  An other example is, again in the medical domain, where the ethnic origin of the subjects has to be taken into account. Then one cannot really build a good model that is not ethnically non-discriminating without schemes of sharing data.

An other aspect that has not been taken account, is the effects of legislation, for example EZU Data governance act that mandates data sharing for data recorder from what the act calls connected devices.

Also, the effects of data sharing based on EU GDP  rights for subjects, not companies, to share their personal data with third parties. These require more complicated data strategy consideration that should be discussed to make the paper valuable for real evaluation of companies data sharing sharing strategies

**Questions:**

I would like the authors to address the points discussed in the weaknesses part of the review, especially redoing their data value analysis based on cases where not including the data from others would lead to discriminative AI models that are generally not acceptable.

Also, even the current analysis should contain the estimation how long time would it take to build an own model (and estimate the cost of being late in the market with AI features) compared to the complications of sharing the data. The equivalent extra time for a attaining a similar product alone could be estimated already with the current model in the manuscript.

**Limitations:**

The libations are discussed in the weakness part - as these are the major weakness, of the otherwise very good paper.

---

> ### Author Rebuttal · Authors · 2023-08-09
>
> Thank you for your valuable and constructive feedback. In the following we address the weaknesses and questions raised in your review.
>
> **Weaknesses**
>
> **The assumption that the collaborative partners share the same data distribution and the data is modeled as iid samples of a common distribution.**
>
> We certainly agree that heterogeneity can be an important aspect in some collaborative learning settings. Please refer to our shared response for a discussion of the numerous challenges and orthogonal incentives arising when considering a heterogeneous setting. In particular, we would like to highlight that since directly incorporating data from other distributions may actually damage the model performance (due to the additional data being too different), heterogeneity yields additional strategic data-sharing considerations (Donahue & Kleinberg, 2021), which are orthogonal to the trade-off considered in this work between improving your model versus risking increased competition.
>
> Additionally, we note that we consider a simple model of heterogeneity in Appendix C.4. We believe that our framework allows for considering even more intricate models of heterogeneity, providing that an appropriate data impact model is formulated. We certainly agree with the reviewer that this is an interesting direction for future work.
>
> Donahue, K. and Kleinberg, J. Model-sharing games: Analyzing federated learning under voluntary participation. In: *AAAI Conference on Artificial Intelligence*, 2021.
>
> **One cannot really build a good model that is not ethnically non-discriminating without schemes of sharing data (in the heterogeneous case).**
>
> We certainly agree that fairness and non-discrimination are important concerns in collaborative learning. In fact, one can explicitly analyze fairness concerns within our framework. To do so, one could consider the impact of fairness constraints on learning outcomes by changing the data impact model and adding some diversity constraints on the action spaces of the firms within their collaboration scheme.
>
> Additionally, we note that already in the homogeneous case, our framework provides a possibility to improve these measures in practice. In particular, even for homogeneous data, the performance of the ML models on rare population groups can be significantly improved by having access to more data from the same distribution (and therefore observing more of the rare group samples).
>
> **Another aspect that has not been taken account, is the effects of legislation, for example EZU Data governance act that mandates data sharing for data recorder from what the act calls connected devices.**
>
> Thank you for the valuable suggestion. One can explicitly incorporate and study the effect of legislation using our framework by constraining the action spaces of the firms in the collaboration scheme (for example, by forcing them to share some parts of data regardless of other data-sharing actions). We see this as an exciting direction for future work, though orthogonal to the current focus of the paper, which analyzes the data-sharing trade-off from the perspective of market incentives only.
>
> Additionally, we believe that our work is aligned with the values and objectives of the EU Data Governance Act since it shows that data sharing can be beneficial even between competing market participants and without the presence of legal requirements to share data.
>
> **The effects of data sharing based on EU GDP rights for subjects, not companies, to share their personal data with third parties … should be discussed to make the paper valuable for real evaluation of companies data-sharing strategies.**
>
> Thank you for the valuable suggestion. We will discuss data ownership aspects in the next version of the manuscript. We expect that this requirement can also be incorporated into our framework. For example, in Section 6, one can implement sharing consent constraints by constraining the firms’ choices of $\lambda$. While, previously, the firms were able to share all data $\lambda \in [0, 1]$, now they can share only data of people who agree with data sharing $\lambda \in [0, \lambda_\text{consent}]$.
>
> **Questions**
>
> **Even the current analysis should contain the estimation how long time would it take to build an own model (and estimate the cost of being late in the market with AI features) compared to the complications of sharing the data. The equivalent extra time for a attaining a similar product alone could be estimated already with the current model in the manuscript.**
>
> Thank you for the valuable suggestion. While training and local data collection costs are certainly interesting to consider, we see these aspects as orthogonal to the focus of our work, which is on the trade-off between receiving access to competitors' data and risking increased market competition. While training can be a bottleneck in some cases (for example, in the context of foundation models), we target the fairly common situation where the lack of sufficient data (and the expensiveness of its collection) are of a much bigger concern.
>
> Additionally, we are unsure how the reviewer suggests to estimate the training costs using our current framework. In order to estimate the effect of being late with the ML model, one needs to model temporal characteristics of the market, as well as costs for local data generation. Both of these aspects are not currently modeled.
>
> We will be happy to expand on our answer, if the reviewer's concern was related to different effects or some other misunderstanding occurred.

---

> > ### Comment · Reviewer_DZoM · 2023-08-15
> >
> > Thank you for the clarifications
> >
> > The consent coefficient is a good addition. It also covers the case of copyrighted data, as there is push (despite of the text and data mining exception) to ask licences for training use of copyrighted works, and companies start to voluntarily accept this.
> >
> > On the training cost case use case, I really meant foundation models,  where it really matters. For training these, gaining as large, and diverse data as possible is crucial.  I think the considerations of this paper are of greatest value in this context, as most of the advances in AI, are currently emerging from transformer like technologies, be in protein folding, automatic coding or image generation etc... The need for large models will overshadow others, not only in use, but also in costs of data.( and in human labelling based grounding), in costs of training, and costs of running. There is an entanglement of these in the deployment that one cannot factorise easily into independent components - like stand alone data strategy, without interfacing the others factors.
> >
> > In this context, there is also other "data" that can be shared and is valuable, like encodings of texts (and images), and the weights of the pre-trained neural networks that can reduce training costs, also it can be reduced by giving out data and asking the recipient to train the model, and get back the trained model. This way of paying training costs with data is actually quite common.
> >
> > Training costs can be estimated by quoting AWS, Azure, Google cloud,...  GPU time pricing.

---

> > > ### Author Response · Authors · 2023-08-15
> > > **Thank you for your response!**
> > >
> > > Thank you for your timely and detailed response!
> > >
> > > We completely agree with the reviewer that foundation models are increasingly prevalent, and their training costs are not negligible. We will be happy to provide a discussion in our manuscript regarding training costs as a possible consideration that may additionally come into play in the context of foundation models.
> > >
> > > While training costs are certainly interesting to model, we note that they give rise to many orthogonal incentives to the trade-off studied in this work. In particular, sharing computation brings up the aspect of fair client and server compensation for training costs, which is a different line of work in the FL incentives literature (Tu et al., 2022).
> > >
> > > Additionally, we see several obstacles in front of the direct modelling of training costs in the context of our problem. Specifically, this will likely require application-specific (and potentially proprietary) information on how these costs are actually incurred. First, it is unclear how the companies will negotiate the training costs splitting. For example, they might split the costs equally among all coalition members, they might split them proportionally to the sizes of their datasets, or, as the reviewer suggested, the central server might bear all the costs, but receive data as compensation. Similarly, there are multiple options how the firms will do inference. They may receive a copy of the end model and use it locally, or they may leave the model at the central server only. If the latter case, it is unclear how they will pay for inference. For instance, they might pay for each query, they might get a quota proportional to their data contribution, or they might auction the server inference time. We will be happy to elaborate on these considerations in the next version of the manuscript.
> > >
> > > Tu, X., Zhu, K., Luong, N. C., Niyato, D., Zhang, Y., and Li, J. Incentive mechanisms for federated learning: From economic and game theoretic perspective. IEEE Transactions on Cognitive Communications and Networking, 2022.

---

### Official Review · Reviewer_4Akt · 2023-07-06

**Soundness:** 3 good
**Presentation:** 3 good
**Contribution:** 3 good
**Rating:** 6
**Confidence:** 3

**Summary:**

This paper explores the dilemma faced by firms when considering strategic data sharing with their competitors. It introduces a framework to investigate the incentives for data sharing and examines its impact on collaboration and profitability. The author discusses the barriers to data sharing, such as privacy concerns, and proposes a market model, data impact model, and collaboration scheme as components of the framework. The findings suggest that reduced competition and harder learning tasks foster collaboration. An illustrative example of a taxi market is provided to demonstrate the concepts discussed. Overall, this study aims to understand how market competition affects collaborative learning incentives and provides insights into the data-sharing trade-off.


**Strengths:**

- The study introduces a novel and comprehensive framework to analyze data sharing between competitors, considering factors such as machine learning model quality's impact on production cost.

- The study investigates the incentives for data sharing and examines the impact of market conditions, product similarities, the complexity of learning task and firm size on collaboration incentives. Albeit it's simply based on a conventional market model and a natural model of data impact grounded in learning theory, the findings are inspiring and the exposition of the results is clear and easy to follow.


**Weaknesses:**

- Since this research is primarily about economic modeling and analysis, it would be great to see a discussion of the various real-world examples in the context rather than just one taxi market example and a oil-market example in the appendix.
- This study uses simulation to examine collaboration incentives among multiple firms, but it would be valuable to discuss the limitations of the simulations and potential biases inherent in such studies.


**Questions:**

- The study mentions the use of a data impact model grounded in learning theory, but it doesn't provide detailed information about the model's validity and discussion about alternative formulations. Is there any other forms of data impact model could also be considered?

- I can see there are plenty extensions provided in the supplementary materials, is there a way to get comparable results for an analog of the problem that consider a sequential game(such as Stackelberg) rather than simultaneous game? This set up may happens in some mega tech companies get the drop on some other small firms and most often the big companies take the advantages.

- The big companies tend to have more bargaining power than small size firms, and the cost functions are often asymmetric. Does Theorem 6.1 still holds when bigger company has different cost function as it is discussed in C.3?


**Limitations:**

The authors didn't discuss the limitations of their theoretical results. This study is mainly theoretical and primarily focused on introducing a novel framework on strategic data sharing dilemma. There are still plenty rooms for in-depth investigations into the data sharing studies.

---

> ### Author Rebuttal · Authors · 2023-08-09
>
> Thank you for your valuable and constructive feedback. In the following we address the weaknesses and questions raised in your review.
>
> **Weaknesses**
>
> **It would be great to see a discussion of the various real-world examples in the context**
>
> Thank you for your valuable suggestion. We will add more examples of industries where such problems arise in the next version of the manuscript. We expect that such problems are particularly relevant for industries such as agriculture, finance, and healthcare (Durrant et al., 2022; FedAI; Rieke et al., 2020), as well as other industries interested in cross-silo FL (Kairouz et al., 2021).
>
> Durrant, A., Markovic, M., Matthews, D., May, D., Enright, J., and Leontidis, G. The role of cross-silo federated learning in facilitating data sharing in the agri-food sector. In: *Computers and Electronics in Agriculture*, 2022.
>
> FedAI. WeBank and Swiss Re signed Cooperation MoU. https://www.fedai.org/news/webank-and-swiss-re-signed-cooperation-mou
>
> Rieke, N., Hancox, J., Li, W., Milletari, F., Roth, H. R., Albarqouni, S., Bakas, S., Galtier, M. N., Landman, B. A., and Maier-Hein, K. The future of digital health with federated learning. In: *NPJ digital medicine*, 2020.
>
> Kairouz, Peter, et al. "Advances and open problems in federated learning." In: *Foundations and Trends in Machine Learning*, 2021.
>
> **It would be valuable to discuss the limitations of the simulations and potential biases inherent in such studies.**
>
> Thank you for pointing this out. We provide a brief discussion here and will update the manuscript's next version accordingly.
>
> First, our simulation solves each instance of the data-sharing game exactly, so we do not anticipate any biases coming from here. Moreover, given that we average over many independent runs (10000), we expect our results to be very close to the true expectation.
>
> Second, we analyze this game to test if the findings from Sections 5 and 6 apply to more complex scenarios involving multiple companies. Therefore, we see Section 7 as further validation of the theoretical findings in the paper rather than a complete characterization of the data-sharing trade-off between multiple firms. Investigating other negotiation schemes between multiple companies will certainly be an interesting direction for future work.
>
> **Questions**
>
> **No detailed information about the data impact model's validity and discussion about alternative formulations. Can other forms of data impact models be considered?**
>
> We refer to the shared response for a discussion on the practical evaluation of the data impact (and market) model. Additionally, please see our response to Reviewer Jw59 for how we expect our results to transfer in case of other data scaling laws suggested by Viering et al. 2022. We also note that Appendices C.3 and C.4 briefly discuss different data impact models. We will make sure to better highlight these aspects in future versions of the manuscript.
>
> Overall, we consider our parametric family to be quite general, at least in the i.i.d. setting, since it covers multiple model-quality convergence rates grounded in standard statistics and ML theory frameworks. In practical applications, we expect the best parametric form of the data impact model to be sensitive to the real-world context.
>
> **Is there a way to get comparable results for an analog of the problem that consider a sequential game (such as Stackelberg) rather than simultaneous game?**
>
> Thank you for the valuable suggestion. There are several ways to include sequential decision-making in our framework. One can assume a sequential competition (for example, according to the Stackelberg model) during the competition phase. Coalition formation could also be made sequential (as in Section 7). If one wants to cover the entry and exit of the firms, they could also iterate our data-sharing game multiple times (similar to super-games in the industrial organization literature). We would be happy to discuss other possibilities if the reviewer provides more details about their setup of interest.
>
> **Does Theorem 6.1 still hold when the bigger company has different cost function as it is discussed in C.3?**
>
> Qualitatively the solution to the bargaining problem should not change, but the numerical constants will be different.

---

> > ### Comment · Reviewer_4Akt · 2023-08-18
> >
> > Thank you for the response. I have a followup comment to Q2 and Q3:
> > - The setup I'm interested in is when some mega tech companies who has different form of costs(maybe lower in \beta_i since they got more resources) than small tech companies and oftenly move first in the market(a Stackelberg game). This may be an overly-detailed setup, but it's more common in real life. And it's intuitively a potential special case for scenarios when companies find it hard to collaborate. It may provide more insights to whole context if it's turned out to be true.

---

> > > ### Author Response · Authors · 2023-08-18
> > > **Analysis of a Stackelberg setup**
> > >
> > > Thank you for your constructive feedback and for the clarification!
> > >
> > > Following your suggestion, we investigated a setup where the competition phase corresponds to a Stackelberg game between two companies. As you suggested, the big company $F_1$ has a better cost function ($\beta_1 > \beta_2$) and more data ($n_1 > n_2$). We consider the full data-sharing negotiation scheme from Section 5.
> > >
> > > Repeating our two-firm analysis (see Appendices A.2, A.3, and A.4) for the Cournot-based Stackelberg game (e.g., Boyer & Moreaux, 1987), where the first firm decides on quantities before the second one, we get the following collaboration criteria:
> > > $$\varPi_{1, \text{ind}}^e \le \varPi_{1, \text{share}}^e \iff \gamma (n_2^{-\beta_2} - (n_1 + n_2)^{-\beta_2}) \le 2 (n_1^{-\beta_1} - (n_1 + n_2)^{-\beta_1}),$$
> > > $$\varPi_{2, \text{ind}}^e \le \varPi_{2, \text{share}}^e \iff \gamma (n_1^{-\beta_1} - (n_1 + n_2)^{-\beta_1}) \le \Bigl(2 - \frac{\gamma^2}{2} \Bigr) (n_2^{-\beta_2} - (n_1 + n_2)^{-\beta_2}).$$
> > >
> > > Here, the first company's incentives to collaborate do not change compared to the Cournot case, while the second company's incentives decrease $\Bigl(2 - \frac{\gamma^2}{2} - \gamma \le 2 - \gamma \Bigr)$. Despite the reduced incentives for the second firm, since $\beta_1 > \beta_2$ and $n_1 > n_2$, the second condition always holds, both in the Stackelberg setup presented here and in the context of Theorem C.1 in Appendix C.3. Therefore, the smaller company will always want to collaborate. Since the incentives of the first firm are unchanged in both cases, there is no change in the likelihood of collaboration compared to the Cournot case. Interestingly, the first firm will have larger profits in the Stackelberg setup, compared to the setup in Appendix C.3, since it could always choose Cournot equilibrium quantities at the first stage of the competition and get the Cournot equilibrium (Anderson & Engers, 1992).
> > >
> > > We hope the reviewer finds this analysis interesting and relevant to their proposed setup. Please let us know if you have any further questions; we will be happy to address them.
> > >
> > > Boyer, M., & Moreaux, M. On Stackelberg equilibria with differentiated products: The critical role of the strategy space. The Journal of Industrial Economics, 217-230, 1987.
> > >
> > > Anderson, S. P., & Engers, M. Stackelberg versus Cournot oligopoly equilibrium. International Journal of Industrial Organization, 10(1), 127-135, 1992.

---

### Official Review · Reviewer_Jw59 · 2023-07-06

**Soundness:** 3 good
**Presentation:** 3 good
**Contribution:** 3 good
**Rating:** 7
**Confidence:** 3

**Summary:**

* This paper analyzes the economic consequences of data sharing between competitors.
* Interaction is modeled as a market:
  * Market has $m$ firms ($F_1,\dots,F_m$), where firm $i$ produces $q_i$ units of good $G_i$ at price $p_i\ge 0$, and quality $v_i$.
  * In addition, there are “outside goods” $\{1,\dots,k\}$ offered at fixed prices $\tilde{p}_l$.
  * Each consumer $j$ optimizes their utility $u^j$ by deciding on consumption of firm goods $q^j\in\mathbb{R}^m_+ $ and outside goods $g^j\in\mathbb{R}_+^k$ under budget constraint $B^j$ - Leading to Firms maximize their expected utility $\mathbb{E}_v[p_i q_i - C_i(q_i,v_i)]$.
  * Different solution concepts are considered: Firms act either by deciding on $q_i$ (Cournot competition), prices $p_i$ (Bertrand competition), or by strategically considering the response of their competitors (Nash equilibrium).
* In section 4, a concrete market model is instantiated: Consumers make decisions according to a quasi-quadratic utility model characterized by a substitutability parameter $\gamma$, and the cost associated with each firm is $C_i(q_i,v_i)=c_i q_i$, where $c_i$ depends on the quality of the machine learning model. Machine learning quality is assumed to affect production costs only.
* For analysis, data is assumed to be homogeneous (data of all firms is sampled independently from the same distribution), and coefficient $c_i$ is assumed to take a concrete power-law parametric form ($c_i = a+b/n^\beta$). Attention is restricted to competition between two firms.
* Theorem 5.1 characterizes the equilibria as a function of the $\gamma$ parameter, and the ratios between the amounts of data collected by the two firms. Theorem 6.1 characterizes the equilibrium in the case of partial data sharing. Finally, a numerical simulation is conducted on a competition setting with more than two firms.


**Strengths:**

* Topic is well motivated.
* Clean presentation, connects contemporary topics in machine learning to classic economic notions in a creative and interesting way.


**Weaknesses:**

* Parametric assumptions for main theorems are not validated against real-world data. Not clear which parameter regimes are likely in practice.
* Data homogeneity assumption may be too conservative - Homogeneity means that datasets collected by all firms are assumed to be sampled independently from the same distribution. This is unlikely to be the case - For instance, in the taxi running example, different operators are likely to observe different data distribution, e.g. because they operate in different parts of town.
* Limitations of the method are not thoroughly discussed.

**Questions:**

* In the proposed model, do the firms have the ability to invest part of their budget in independent data collection? (e.g by conducting surveys to improve training set quality, or buying data from an external provider which is not a direct competitor). If the market model does not include this possibility, what would be the consequences of considering it?
* What are typical values of the constants $a$,$b$,$\gamma$ in real-world systems, and why?
* L224: “We assume that $b/(1-a)$ is small enough...” - Which values of $b/(1-a)$ are small enough for the lemmas to hold? Are they realistic in real-world systems?
* How would the results be different if the cost $c_i$ had a different parametric form? For example, using similar arguments to the ones presented in the paper, one could consider costs that relate to the parametric forms described in Viering et al. 2022 (Table 1).


**Limitations:**

Limitations are not thoroughly discussed. In particular, I feel it would be helpful to provide an indication of where we expect the main assumptions in this paper to hold, and discuss the consequences of making wrong assumptions.

---

> ### Author Rebuttal · Authors · 2023-08-09
>
> Thank you for your valuable and constructive feedback. In the following we address the weaknesses and questions raised in your review.
>
> **Weaknesses**
>
> **Parametric assumptions for main theorems are not validated against real-world data. Not clear which parameter regimes are likely in practice.**
>
> Thank you for your feedback. We show how one can reason about parameter values in our response in the "Questions" section below. As we note in the general response, a systematic analysis and validation of a real-world market model and the most exact data impact model are probably outside our submission's scope. Please refer to the main response for our motivation.
>
> **Data homogeneity assumption may be too conservative**
>
> We certainly agree that heterogeneity may be an important aspect in some settings. Please refer to our shared response for a discussion of the numerous challenges and orthogonal incentives arising when considering a heterogeneous setting. Additionally, we note that a simple model of heterogeneity is considered in Appendix C.4. Our framework allows even more intricate models of heterogeneity, providing that an appropriate data impact model is formulated.
>
>
> **Questions**
>
> **In the proposed model, do the firms have the ability to invest part of their budget in independent data collection? … If the market model does not include this possibility, what would be the consequences of considering it?**
>
> Thank you for the valuable suggestion. One can incorporate this possibility within our framework by adding a data-collection stage of the game before the data-sharing stage. Intuitively, it should incentivize firms with small amounts of data to gather more of it to be accepted into desirable coalitions. We expect the results of Theorem 5.1 qualitatively stay the same, but this mechanism may lower the threshold for collaboration.
>
> **What are typical values of the constants $a, b$, $\gamma$ in real-world systems, and why?**
>
> While our market model is widely used in economic theory, one typically uses domain-specific models to describe real-world systems. As we pointed out in the general response, we hope that our three-component decomposition of the data-sharing problem will help practitioners to leverage their situation-specific knowledge of their market and ML models to formulate an appropriate application-specific model.
>
> That said, the parameters in our manuscript do have natural interpretations that can inform relevant magnitudes based on economic intuition. In particular, $\gamma$ should be positive (since we consider competing firms) but not very close to $1$ (since firms usually do not produce identical products). The parameter $a$ should correspond to the costs of the firms with a perfect ML model, and $a + b$ should correspond to the costs of the firms with a very bad model. Since $1$ should correspond to the largest possible price for a product (when there is only an infinitesimally small amount of this good on the market), the ratio $\frac{1}{a}$ should roughly correspond to the ratio of the highest possible price of the product (like during supply shortage) to the lowest possible price. For example, this multiplier for the natural gas in the EU during 2020–2022 was around 50 (FRED), implying that $a$ was around $\frac{1}{100}$. As for the ratio of $\frac{b}{a+b}$, we could use consulting firms' data about the impact of machine learning on costs. In Espel et al. (2020), this ratio was around $\frac{1}{5}$, yielding values of $b$ around $\frac{1}{500}$.
>
> FRED. Global price of Natural gas, EU. https://fred.stlouisfed.org/series/PNGASEUUSDM
>
> Espel, P., Herbener, M., Rupprecht, F., Schröpfer, C., and Venus, A. How industrial companies can cut their indirect costs—fast. https://www.mckinsey.com/industries/automotive-and-assembly/our-insights/how-industrial-companies-can-cut-their-indirect-costs-fast, McKinsey, 2020.
>
> **Which values of $b / (1 - a)$ are small enough for the lemmas to hold? Are they realistic in real-world systems?**
>
> We provided these restrictions in the proofs of our lemmas (Appendices A.3, L490 and A.4, L504). For example, in the Cournot case, the restriction would follow from the following property: $1 - a > m b$. Given our discussion of magnitudes in the previous paragraph, we expect this property to hold.
>
> Intuitively, in our market model, the ratio being small is equivalent to the assumption that firms do not exit the market during the competition stage. While modeling firms' entry into and exit from the market is certainly interesting, we see this mechanism as complementary to our focus on the trade-off described in this work between improving your model versus risking increased competition.
>
> **How would the results be different if the cost had a different parametric form? For example, costs that relate to the parametric forms described in Viering et al. 2022 (Table 1).**
>
> We do not expect the results to change qualitatively. However, instead of a threshold for the ratio between the number of data points, we will have another disparity function between them (for example, difference), and the notion of task simplicity would be different.

---

> > ### Comment · Reviewer_Jw59 · 2023-08-15
> >
> > Thank you for the thorough and helpful response! In particular, I appreciate the order-of-magnitude estimation of model parameters, and I believe that such grounding significantly strengthens the presented results. Given the suggestions you made for improvement, I view the paper as a step in the right direction, and I believe it can facilitate fruitful discussions within the community. I'm increasing my rating to 7 (Accept).

---

> > > ### Author Response · Authors · 2023-08-15
> > > **Thank you for your response!**
> > >
> > > Thank you for your timely response! We appreciate your constructive and positive feedback, which we will incorporate in the next version of the manuscript. In particular, we will include the discussion on the model parameters $a, b, \gamma$.

---

### Author Rebuttal · Authors · 2023-08-09

We thank all reviewers for their valuable and constructive feedback. Below we provide a general response to several recurring comments. We also offer individual answers for all remaining questions.

**Real-world evaluation of the market and data impact models.**

Multiple reviewers brought up the importance of real-world evaluation of the considered market and data impact models studied in Sections 4-7. While we recognize the significance of this concern, we see the core contributions of our paper in a general framework for the data-sharing trade-off and general qualitative insights into this problem. Our approach is more theoretical in nature and does not aim to provide a fully-realistic model that can directly inform practitioners. We will make sure to further clarify the scope of our work in the next version of the manuscript.

We opted for a theoretical approach for the following reasons.

First, we do not anticipate a one-size-fits-all model to cover all data-sharing scenarios, considering the diversity of relevant industries (e.g., online platforms, finance, agriculture, healthcare) and machine learning tasks (e.g., mean estimation, regression, deep learning). In practice, designing the most appropriate models will likely be an application-dependent problem, and we hope that our three-component decomposition of the data-sharing problem will help practitioners to leverage their situation-specific knowledge of their market and ML models.

Moreover, both economics and machine learning lack a definitive theory to predict parametric forms of market and data impact models for particular situations. In economics, market modeling is more often art than science, and parametric models (e.g., Berry et al., 1995; Nevo, 2001) require the scope of one economic article to explain and justify (since they require us to model counterfactuals). In machine learning, the generalization of deep models, especially for non-i.i.d. data, remains poorly understood.

Third, already with the basic market and data impact models, a complete analytical solution to the problem in Sections 6 and 7 remains elusive (although we provided various qualitative insights about them). With more nuanced market and data impact models, we expect to get only numerical results without rigorous theoretical characterizations. At the same time, we used a market model widely adopted in the theoretical literature (Choné & Linnemer, 2019) and a data impact model motivated by established theoretical frameworks in machine learning (Tsybakov, 2004). Given the wide adoption of these parametric families, we hope that our results will be useful and qualitatively valid in real-world data-sharing settings.

**The assumption of homogeneous data**

We certainly agree that heterogeneity is an important concern in collaboration learning. In case of significant heterogeneity, our framework will only require appropriate changes to the data impact model. We expect that designing such a model is a significant challenge orthogonal to the focus of our work (see, for example, Gulrajani & Lopez-Paz, 2020). Moreover, in some cases, additional data from a different distribution may damage model performance, yielding additional data-sharing considerations (Donahue & Kleinberg, 2021). Given these concerns, we decided to focus on the homogeneous case and also discuss a simple model of heterogeneous learning in Appendix C.4. We certainly agree the analysis of heterogeneity is an exciting direction for future work.

Berry, S., Levinsohn, J., and Pakes, A. Automobile Prices in Market Equilibrium. In: *Econometrica: Journal of the Econometric Society*, 1995.

Choné, P. and Linnemer, L. The quasilinear quadratic utility model: An overview. *CESifo Working Paper*, 2019.

Donahue, K. and Kleinberg, J. Model-sharing games: Analyzing federated learning under voluntary participation. In: *AAAI Conference on Artificial Intelligence*, 2021.

Gulrajani, I. and Lopez-Paz, D. In Search of Lost Domain Generalization. In: *International Conference on Learning Representations (ICLR)*, 2020.

Nevo, A. Measuring market power in the ready‐to‐eat cereal industry. In: *Econometrica*, 2001.

Tsybakov, A. B. Optimal aggregation of classifiers in statistical learning. In: *The Annals of Statistics*, 2004.

---

### Decision · Program_Chairs · 2023-09-21

**Decision:**

Accept (poster)

**Comment:**

This paper presents a general framework for analyzing data-sharing trade-off and analysis of an instance of this framework. Some interesting insights can be drawn from the analysis. We hope the reviews and feedbacks during the discussions could be helpful for the authors to better prepare the final version of this paper.